



# From air quality to climate; Impact of aerosol sources on optical properties at urban, regional and continental levels in the north-western Mediterranean

Marina Ealo[1][2], Andrés Alastuey[1], Noemí Pérez[1], Anna Ripoll[1], Xavier Querol[1], Marco Pandolfi[1].

[1]{Institute of Environmental Assessment and Water Research (IDAEA-CSIC), Barcelona, Spain}.

[2]{Group of Meteorology, Department of Applied Physics, Faculty of Physics, University of Barcelona, Spain}.

Correspondence to: Marina Ealo (marina.ealo@idaea.csic.es)

## Abstract

Further research is needed to reduce the existing uncertainties on the effect that specific aerosol sources have on radiative forcing, thus supporting the assessment of future mitigation strategies which should be focused on both air quality and climate, and not acting separately. This study presents a new approach aimed at quantifying the mass scattering and absorption efficiencies (MSE and MAE) of different aerosol sources at urban (Barcelona-BCN), regional (Montseny-MSY) and remote (Montsec-MSA) background sites in the northwestern (NW) Mediterranean. An analysis of source apportionment to the measured scattering and absorption coefficients was performed by means of a multilinear regression (MLR) model during 2010-2014 at BCN and MSY and during 2011-2014 at MSA. The source contributions to $PM_{10}$ mass, identified by means of the Positive Matrix Factorization (PMF) model, were used as dependent variables in the MLR model in order to take into account the internal mixing state of atmospheric aerosols.

Seven aerosol sources were obtained at MSA and MSY and 8 sources at BCN. *Mineral*, *Aged marine*, *Ammonium sulfate*, *Ammonium nitrate* and *V-Ni bearing* sources were common at the three sites. *Traffic*, *Industrial/metallurgy* and *Road-resuspension* were isolated at BCN, whereas *Industrial/Traffic* and *Aged organics* were solely identified at MSY and MSA. The highest MSE were found for *Ammonium sulfate* (4.5 and 10.7 $m^2 g^{-1}$), *Ammonium nitrate* (8.8 and 7.8 $m^2 g^{-1}$) and *V-Ni* (8 and 3.5 $m^2 g^{-1}$) at MSY and MSA respectively, dominating the scattering throughout the year with marked seasonal trends. *V-Ni bearing*, originated mainly from shipping emissions, simultaneously contributed to both scattering and absorption being the second most efficient absorptive source in BCN (0.9 $m^2 g^{-1}$). The *Traffic* source at BCN and the equivalent *Industrial/Traffic* at MSY and MSA mainly governed the light absorption and exhibited the highest MAE (1.7, 0.9 and 0.2 $m^2 g^{-1}$, respectively). Sources predominantly composed by fine and relatively dark particles such as *Industrial/Traffic, Aged organics* and *V-Ni* were simultaneously characterized with low single scattering albedo (SSA) and high scattering Angstrom exponent (SAE). Conversely, *Mineral* and *Aged marine* showed the highest SAE and the lowest SSA, being scattering the dominant process in the light extinction. The good agreement between modeled and measured optical properties allowed for the reconstruction of scattering, absorption and SSA time series by means of the PMF-MLR technique for the period 2004-2014 at MSY. Significant decreasing trends were found for the modeled scattering and absorption (-4.6 and -4.1 % $y^{-1}$)




coefficients. Interestingly, the observed reduction in the SSA (-0.11 % $y^{-1}$) might suggests a less effectiveness of the air quality strategies focused on reducing pollutants containing black carbon (BC) particles, which highly contribute to light absorption and thus climate warming.



# 1. Introduction

Further research on the sources of atmospheric pollutants and their potential effect on climate forcing is needed in order to assess future mitigation strategies. The air quality abatement strategies adopted in the recent years have resulted in a decrease of anthropogenic pollutants in Europe (EEA, 2013; Barmpadimos et al., 2012; Querol et al., 2014; Pandolfi et al., 2016).

However, the control of pollutant emissions is currently conflicting involving a trade-off between the impacts on environmental health and the Earth's climate, and therefore current mitigation strategies could increase climate warming while improving air quality (Shindell et al., 2012). Thus, precise measurements of aerosol properties are required to further deepening in the relationship existing between aerosol optical and chemical properties, in order to better understand the link air quality-climate and to reduce the current uncertainties on the effects that aerosols have on radiative forcing (IPCC 2007,

10 2013).

Atmospheric aerosols affect the Earth's climate through the direct scattering and absorption of solar radiation but also by indirect processes acting as cloud condensation nuclei (IPCC, 2007). Most aerosol components (mainly sulfate, nitrate, organics and mineral matter) scatter the sunlight causing a net cooling at the top of the atmosphere (TOA); conversely other particles such as black carbon (BC) absorb solar radiation in the whole visible spectrum, thus warming the atmosphere and

causing a net warming at TOA (Jacobson, 2001a; Ramanathan and Carmichael, 2008). On the global scale, aerosols are estimated to cool the Earth system (Chen et al., 2011; IPCC, 2013). Assessing the role of aerosols on climate forcing often requires reducing their physicochemical properties to a set of parameters that describe their optical properties, such as the mass scattering and absorption efficiencies (MSE and MAE respectively) (Hand and Malm, 2007). These are key parameters that relate the aerosol scattering and absorption coefficients with mass concentration of specific species, thus linking the

aerosol chemical and optical properties of climate relevant species. MSE and MAE are intensive optical parameters depending on intrinsic aerosol properties such as particle effective radius, particle mass density or refractive index, and they are very useful to better parameterize the aerosols direct radiative effect in atmospheric models (Seinfeld and Pandis, 1998; Hand and Malm, 2007); and therefore to quantify the potential of aerosols for cooling or warming the Earth-atmosphere system.

However, a thorough quantification of the direct and indirect aerosol effects on the Earth's radiative budget is difficult to achieve, due to the high spatial and temporal variability along with the large differences in aerosol composition and size (Zieger et al., 2012). Aerosols originate from a huge variety of sources (both natural and anthropogenic) and formation and transformation processes, leading to a widely variation in type, composition and mixing state (i.e. Haywood et al., 1999; Andrews et al., 2011; Bond et al., 2013). Moreover, given their short lifetime (from days to weeks) and the number of

sources and sinks, the broad spatial and temporal variability of aerosol properties is extremely high and this results in a radiative forcing from local to global scales (Collaud Coen et al., 2013).

The determination of MSE and MAE for specific aerosol components has been subject of research in the last few years. Several studies focused on determining the MAE of black carbon (BC) particles, as the ratio between aerosol absorption





coefficient and elemental carbon (EC) concentrations obtained by means of thermo-optical analysis on filter samples (i.e. Bond et al., 2013; Reche et al., 2011; Pandolfi et al., 2011, 2014). In some cases, the MAE of BC has been observed to change depending on the degree of the internal mixing of BC with non-absorbing material such as sulfate and organic compounds (Jacobson, 2001b; Moffet and Prather, 2009; Ramana et al., 2010; Zanatta et al., 2016). The MSE of different

chemical aerosol components has been also broadly reported for many locations (Vrekoussis et al., 2005; Titos et al., 2012; Cheng et al., 2015 and references therein). An example is the study performed by the IMPROVE (Interagency Monitoring of Protected Visual Environments) program which has been taken as a reference for reporting mass extinction (scattering + absorption) efficiencies (MEE) depending on particle composition (Hand and Malm, 2006 and 2007). Global MSE were reported for dry ammonium sulfate [$(NH_4)_2SO_4$], ammonium nitrate [$NH_4NO_3$], organic matter (OM), soil dust and sea salt

from long term measurements (from 1990 to 2007) in U.S National Parks. The IMPROVE algorithm is based on a multilinear regression (MLR) of chemical species concentration to the measured aerosol scattering or extinction coefficients, thus assuming an externally mixed aerosol. Ammonium sulfate and ammonium nitrate were stoichiometrically calculated, the mineral matter and sea salt are obtained from assumed forms of their chemical tracers (Al, Si, Ti, Ca and Fe, and Na and Cl respectively) and OM is usually estimated from organic carbon (OC) measurements using convenient OM-to-OC ratios

(Turpin et al., 2000). However, a unique light absorbing carbonaceous material (EC) is considered for the MAE estimation. Opportune correction factors for some chemical species are included in the IMPROVE formula in order to account for the effect of relative humidity in the scattering enhancement. Once the MEE from the IMPROVE algorithm are obtained, the aerosol extinction coefficient can be reconstructed using the concentration of the main chemical components of particulate matter (PM) and the corresponding MEE. A broad variability for MSE and MAE coefficients has been found in the

literature. Differences arise from the different sampling and analysis methods used for determining chemical and optical properties, but also because the mixing state of atmospheric aerosols which broadly vary from one site to another. Therefore, coefficients obtained in the IMPROVE equation might not be used in other emplacements since MSE and MAE are site dependent and unique coefficients. Revised algorithms of the IMPROVE equation have been published in literature aimed at reducing the bias on the predicted values, accounting for a 25% overestimate in the scattering coefficient (Ryan et al., 2005;

Pitchford et al., 2007). For example Ryan et al. (2005) reconstructed light extinction using a concentration power law model, that assumes concentration varying mass scattering efficiencies, resulting in a 2% overestimation for the measured scattering coefficients. This method allows obtaining accurate predicted values for the reconstruction of long-term series of optical properties; however, it cannot be used for a reliable calculation of MSE and MAE. Additionally, the internal mixing state of atmospheric aerosols is not considered in none of these methods given that each major chemical specie is treated separately

from the other, and probably leading to biases due to aerosols not considered in these studies.

In the present study a different approach of the MLR method was investigated using PM source contributions obtained by means of the PMF (Positive Matrix Factorization) model as a substitute for chemical species. An important characteristic of the PMF sources is that these take into account the internal mixing of atmospheric aerosols as evidenced by their chemical profiles, which are constituted by the main tracers defining the source but are also enriched in other chemical compounds.





With this aim, a source apportionment to $PM_{10}$ was performed at Montsec (MSA) by means of the PMF model during the period 2010-2014. PM source contributions obtained at the Barcelona (BCN) and Montseny (MSY) sites using a unique chemical data set of 11 years (2004-2014) were also considered in this study (Pandolfi et al., 2016). From one side receptor models such as PMF are powerful and widely used techniques to design air quality strategies (i.e. Viana et al., 2008; Belis et al., 2013) due to the capability of these models to identify key pollutant emission sources and calculate their contributions to the measured PM mass. Actually, Belis et al. (2014) proposed the use of harmonized methods for source apportioning across Europe given the relevant information these models provide for the development and assessment of pollution source abatement measures. On the other side, the regression model (MLR) of source contributions to scattering and absorption coefficients allows to quantify the potential of these sources for scattering or absorbing the light, and therefore directly linking the air quality and climate effects of airborne PM.

The PMF+MLR technique was performed for the aerosol scattering and absorption coefficients measured at three sampling sites aiming to calculate the MSE and MAE of the different sources identified at urban, regional and remote environments in the NE of Spain since 2010. Furthermore, source contributions obtained from the PMF model and the corresponding MSE and MAE computed by means of the MLR (when optical measurements were available) were used for the reconstruction of the aerosol scattering and absorption coefficients over the last eleven years. This methodology allowed to study long term trends of optical parameters at MSY, which are extremely relevant for the detection of changes in atmospheric composition depending on changes in natural or anthropogenic emissions, atmospheric processes and sinks (Collaud Coen et al., 2013). A relevant outcome of this new approach is the chance to study the effects that air quality mitigation strategies are having on climate in the area under study.

## 2. Methodology

### 2.1 Sampling sites and meteorology

The Western Mediterranean Basin (WMB) is characterized by warm summers and temperate winters with irregular precipitation rates throughout the year. In winter the location of the Azores high-pressure system favours the entry of Atlantic advections that clears the atmosphere out of pollutants. In summer, atmospheric dynamics coupled to local orography result in local/regional circulations with the consequent accumulation of pollutants (Millán et al., 1997). Recirculation and aging of pollutants is favoured by weak gradient atmospheric conditions, scarce precipitation and continuous exposure to solar radiation driving photochemical reactions (Rodríguez et al., 2002; Pérez et al., 2004). Additionally, large mineral dust contributions from Saharan dust events may cause exceedances of the air quality standards (Querol et al., 1998 and 2001; Escudero et al., 2007). The conjunction of all these processes surrounding the WMB, lead to a radiative forcing among the highest in the word (Jacobson, 2001a).

PM chemical and optical measurements were performed at three sampling sites located in the NE Spain (Fig. 1):



The large coastal Barcelona urban area (BCN; 41°23′N, 02°6′E, 80 m a.s.l.) is one of the most populated areas in the Northwestern (NW) Mediterranean, resulting in a very high road traffic density. Additionally, the metropolitan area is surrounded by a broad industrial sector and has one of the main harbors in the Mediterranean Basin, with a large number of cruise ships (Pey et al., 2013). All these anthropogenic emission sources contribute to the largest atmospheric pollution in the area causing important consequences on air quality (Querol et al., 2001; Pey et al., 2008; Pérez et al., 2008; Amato et al., 2009; Reche et al., 2011; Dall'Osto et al., 2013).

The Montseny regional background station (MSY; 41°19′N, 02°21′E, 720 m a.s.l.) is located in the Montseny natural park in a densely forested area, 50km to the N–NE of the Barcelona urban area and 25km from the Mediterranean coast. Despite the site is far enough from the industrialized and populated Barcelona metropolitan region, it can be affected by anthropogenic emissions transported to inland areas (Pérez et al., 2008).

The Montsec continental background site is a remote high altitude emplacement (MSA; 42°3′N, 0°44′E, 1570 m a.s.l.) placed in the southern side of the Pre-Pyreness at the Montsec d'Ares mountain range, located 140 km to the NW of Barcelona and 140 km to the WNW of MSY. Despite the high-altitude location and the frequent free troposphere conditions during the cold season, the station can be slightly influenced by anthropogenic emissions during the warmer period, when it is positioned within the planetary boundary layer (PBL), (Ripoll et al., 2014).

The three sites are involved in the Catalonian air quality monitoring network. Additionally, the MSY and MSA stations form part of the ACTRIS (Aerosol, Clouds and Trace gases Research InfraStructure) and GAW (Global Atmosphere Watch) networks, and then aerosol optical measurements were performed following the standards required by these networks. Further information characterizing physical, chemical and optical properties of atmospheric aerosols detailing the prevailing atmospheric dynamics at the three stations can be found in: Querol et al., (2001); Pey et al. (2009 and 2010); Reche et al. (2011); Pandolfi et al. (2011 and 2014); Cusack et al. (2012); Ripoll et al. (2015) and Ealo et al. (2016).

### 2.2 Measurements and instrumentation

Aerosol light scattering coefficients were measured every 5 min at three wavelengths (450, 525 and 635 nm) with a LED-based integrating nephelometer (model Aurora 3000, ECOTECH Pty, Ltd, Knoxfield, Australia). Scattering measurements were obtained for MSY since 2010 using a $PM_{10}$ cut-off inlet. Measurements at MSA were carried out using a $PM_{2.5}$ cut-off inlet from 2011 until March 2014, and then replaced with a $PM_{10}$ cut-off inlet. Scattering measurements at BCN were not available. Calibration of the two nephelometers was performed three times per year using $CO_2$ as span gas, while zero adjusts were performed once per day using internally filtered particle free air. The relative humidity (RH) threshold was set following the ACTRIS recommendations (RH<40%). Scattering measurements were corrected for truncation due to non-ideal detection of scattered radiation following the procedure described in Müller et al. (2011b).




Aerosol light absorption coefficient at 637 nm (Müller et al., 2011a) was measured at 1 min resolution with a Multi Angle Absorption Photometer (MAAP, model 5012, Thermo), operated in the heated sampling mode and connected to a $PM_{10}$ cut-off inlet at BCN, MSY and MSA.

Gravimetric $PM_{10}$ mass concentrations were determined by standard gravimetric procedures, according to UNE-EN 12341, 1999 protocol. Samples were collected every 3/4 days on 150 mm quartz micro-fiber filters (Pallflex 2500 QAT-UP and Whatman QMH) using high-volume samplers (DIGITEL DH80 and/or MCV CAV-A/MSb at 30 $m^3$ $h^{-1}$).

Chemical off-line filter analyses were carried out at the three sites following the procedure proposed by Querol et al. (2001). A quarter of the filter was acid digested ($HNO_3$:HF:$HClO_4$). The resulting solution was analyzed by Inductively Coupled Plasma Atomic Emission Spectroscopy (ICP-AES; IRIS Advantage TJA Solutions THERMO) for the determination of major elements (Al, Ca, Fe, K, Na, Mg, S, Ti, P) and by Inductively Coupled Plasma Mass Spectrometry (ICP-MS; X Series II, THERMO) for the trace elements (Li, Ti, V, Cr, Mn, Co, Ni, Cu, Zn, As, Se, Rb, Sr, Cd, Sn, Sb, Ba, rare earths, Pb, Bi, Th, U). In order to examine the accuracy of the acid digestion, a few milligrams of the reference material NIST 1633b were added to a quarter of the blank filter. Another quarter of each filter was water extracted to determine soluble anions. The nitrate, sulfate and chloride concentrations were resolved by Ion Chromatography (IC) and the ammonium concentrations with an ion selective electrode (MODEL 710 A+, THERMO Orion). OC and EC concentrations were determined by a thermal-optical carbon analyzer (SUNSET) following the EUSAAR2 thermal protocol (Cavalli et al., 2010). Blank filters were analyzed together with the samples, and concentrations were subtracted from those found in the samples in order to calculate the ambient concentrations. Chemical filter analysis presented in this study ranges between the period 2004-2014 at BCN and MSY, and the period 2010-2014 at MSA. Optical measurements were considered for the periods 2010-2014 at BCN and MSY and 2011-2014 at MSA.

## 2.4 Positive Matrix Factorization model (PMF)

The positive matrix factorization model (PMFv5.0, EPA) was individually applied on the daily chemical speciated data collected at BCN, MSY and MSA for source identification and apportionment in $PM_{10}$. Source contributions obtained for BCN and MSY can be found in Pandolfi et al. (2016), whereas sources identified at MSA will be presented in this study. Detailed information describing the PMF model can be found in literature (Paatero and Tapper, 1994; Paatero, 1997; Paatero and Hopke, 2003; Paatero et al., 2005).

Briefly, the PMF model is a factor analytical tool based on the weighted least-squares method, which reduces the dimension of the input matrix (i.e. the daily chemical speciated data) in a limited number of factors (or sources). Calculation of individual uncertainties and detection limits were based on the approach by Escrig et al. (2009) and Amato et al. (2009), where both the analytical uncertainties and the standard deviations of species concentrations in the blank filters were considered in the uncertainties calculation. This procedure provides a criterion to separate the species which retain a significant signal from the ones dominated by noise, based on the signal-to-noise S/N ratio defined by Paatero and Hopke




(2003). Species with S/N<2 are generally defined as weak variables. In order to avoid any bias in the PMF results the data matrix was uncensored, i.e. negative, zero and below detection limit values were included in the analyses.

The PMF was run in robust mode (Paatero, 1997) and rotational ambiguity was handled by means of the FPEAK parameter (Paatero et al., 2005). The final number of sources was selected based on several criteria: investigating the variation of the

objective function Q (defined as the ratio between residuals and errors in each data value) depending on the number of sources (i.e. Paatero et al., 2002), studying the physical meaningfulness of factor profiles and contributions, and analyzing the scaled residuals and the G space plots.

## 2.5 Multilinear regression model

The Multilinear regression (MLR) method has been previously used in order to determine the scattering and extinction efficiencies (MSE and MEE) of chemical species, that correspond to the linear regression coefficients of the model (White et al.,1986; de Vasconcelos et al., 2001; Hand and Malm, 2007). In the present study, the PMF source contributions were considered as dependent variables and the measured scattering (Sc) and absorption (Abs) coefficients were treated as independent ones. As a result, a unique MSE and MAE can be obtained for each source. The partial scattering and

absorption contribution for each source can be computed as the product between the source concentration and the corresponding MSE/MAE, and then total aerosol light scattering and absorption coefficients can be modeled (Eq. 1 and 2).

$$\text{Sc}^{\lambda}_{\text{PM}_{10}} = \sum\left(MSE^{\lambda}_{source}\cdot[source]\right) \qquad\qquad \text{(Equation 1)}$$

$$\text{Abs}^{\lambda}_{\text{PM}_{10}} = \sum\left(MAE^{\lambda}_{source}\cdot[source]\right) \qquad\qquad \text{(Equation 2)}$$

The apportionment of scattering by more than one species to the total scattering depends on the assumption of the internal or external mixing state of atmospheric aerosols, as already noted previous studies using chemical species assuming an externally mixed aerosol (White, 1986). This study presents a different approach using the $PM_{10}$ source contributions as dependent variables in the MLR instead of chemical species concentrations for computing MSE and MAE. Thus, the resulting regression coefficients represent the scattering and absorption efficiencies of mixed aerosol modes given that the

sources from PMF take into account the possible internal mixing. The assumption of an internal mixing among the different chemical species that form a single variable in the regression equation reduces some of the collinearity issues and makes the regression coefficients less sensitive to data uncertainties (Hand and Malm, 2007). Data collinearity due to meteorological fluctuations is also considered using the source contributions identified by PMF and then derived MSE and MAE are not biased by dispersion patterns. Actually, the matrix correlation showed in Fig. S1 demonstrates the low correlation presented

between pairs of aerosol sources identified at MSA. Moreover, a better model perform is achieved taking into account the full chemical speciation used in the PMF for source apportioning, given that the MLR method assumes that all the species contributing to scattering and absorption are included in the equation.





It is remarkable that differences in the sampling conditions (RH, size cut) or chemical analysis methods influence the resulting efficiencies obtained for different emplacements. In this study, scattering RH was controlled below 40% preventing the hygroscopic growth of the particles, which could lead to an enhancement in the scattering efficiency. An enhancement of the scattering or absorption efficiencies is also related with the known bias in the MLR. This method tends to give more weight to those variables that are more accurately measured (such as sulfate), and conversely underestimate coefficients for those species with larger uncertainty (organic matter) (White and Macias, 1987; Hand and Malm, 2007). In the present study a comparison between modeled and measured coefficients was performed using the fractional bias (FB) described in equation 3 (Ryan et al., 2005). Where $Sc^{sim}$ is the modeled scattering coefficient and $Sc$ is the measured value, for each daily data point.

$$FB = \frac{Sc^{sim} - Sc}{Sc}$$ (Equation 3)

A total of 303, 379 and 503 data points were used in the MLR analysis for source apportioning to absorption at MSA, MSY and BCN, respectively. Whereas 222 and 307 data points were considered for MSE and MAE calculation at MSA and MSY, ensuring that the number of samples is large enough to provide stable results.

**2.6 Statistical tests for trends study**

The Theil-sen slope estimate (TS) (Theil 1950; Sen 1968) is a non-parametric test which was investigated for the monthly averages of scattering, absorption and SSA in order to test for the occurrence of a non-null slope in the data series during the period 2004-2014 at MSY. The total and annual reduction of these optical parameters was investigated using bootstrap resambling for the monthly deseasonalized time series, reducing the possible influence of outliers on trend estimates and obtaining robust slope p-values.

A multi-exponential fit developed within the Task Force on Measurements and Modelling (TFMM) by the Meteorological Synthesizing Centre – East (MSC-E) group (Shatalov et al., 2015), aimed at studying temporal trends of air pollution in the multi-exponential form, was used for representing the decomposed modeled monthly temporal series in: main component, seasonal component and residual component. Additionally, this technique allowed us to estimate the non-linearity (NL) parameter for the trends. An NL of 10% was used as threshold to define a linear trend (NL<10%).

**3. Results**

**3.1 Source profiles and contributions to PM$_{10}$**

Seven aerosol sources were identified in the PM$_{10}$ fraction by performing a PMF model at the MSA station during the period 2010-2014. The source chemical profiles and contributions to the measured PM$_{10}$ mass are shown in Fig. 2 and Table 1. Average absolute and relative source contributions obtained for BCN and MSY during the period 2004-2014, also





considered in this study (Table 1), have been previously quantified by Pandolfi et al. (2016). Higher $PM_{10}$ average concentrations were found at the BCN urban station, followed by the regional (MSY) and remote (MSA) background sites (34.0, 16.7 and 9.6 $\mu g\ m^{-3}$, respectively).

On average, the most abundant sources contributing to $PM_{10}$ levels at MSA were *Aged organics*, followed by *Mineral*,
*Industrial/Traffic*, *Aged marine*, *Ammonium sulfate*, *V-Ni bearing* and *Ammonium nitrate*. *Aged organics* is mainly traced by OC and EC with maxima in summer, pointing to a large contribution from biogenic emission sources and accounting for 2.8 $\mu g\ m^{-3}$ (29.4%) of the $PM_{10}$ load. The EC internally mixed within this source suggests a clear anthropogenic contribution despite the remote environment of MSA. However, *Aged organics* is considered to be dominated by secondary organic aerosols (SOA) arising from biogenic volatile organic compounds (VOCs) due to the predominance of OC in the chemical
profile. Furthermore, it should also be considered the higher summer VOCs oxidative potential occurring in the Mediterranean, due to both higher insolation and tropospheric ozone concentration, which enhance the OC concentrations (Fuzzi et al., 2006). *Mineral*, which is traced by typical crustal elements such as Al, Ca, Mg, Fe, Ti, Rb and Sr, is related to both Saharan dust events and regional/local mineral contribution accounting for an averaged $PM_{10}$ load of 2.27 $\mu g\ m^{-3}$ (23.6%). The *Industrial/Traffic* source is primarily traced by Pb, Zn, As, Sb, Cu and Ni and contributed 1.09 $\mu g\ m^{-3}$ (11.3%).
*Aged marine* is traced mainly by Na and Cl, and in a minor proportion by Mg, $SO_4^{2-}$ and $NO_3^-$, contributing 1.08 $\mu g\ m^{-3}$ (11.1%). *Ammonium sulfate* is a secondary inorganic source traced mainly by $SO_4^{2-}$ and $NH_4^+$, contributing 0.87 $\mu g\ m^{-3}$ (9.0%). *V-Ni bearing* source is traced mainly by V, Ni and $SO_4^{2-}$ and represents the direct emissions from heavy oil combustion, mainly shipping in the study area, contributing 0.79 $\mu g\ m^{-3}$ (8.2%) to the total $PM_{10}$. Finally *Ammonium nitrate*, also secondary inorganic source, is traced by $NO_3^-$ and $NH_4^+$ and enriched in EC accounting for 0.72 $\mu g\ m^{-3}$ (7.5%) of the
$PM_{10}$ load.

As summarized in Table 1, some common sources were identified at the three stations: *Mineral, Aged marine, Ammonium nitrate*, *Ammonium sulfate*, *V-Ni* and *Traffic* (at BCN) or the equivalent *Industrial/Traffic* (at MSY and MSA). At BCN, sources traced by pollutants from anthropogenic activities are related with fresh emissions from the metropolitan area (*Traffic*, *Road resuspension*), the surrounding industrial zone (*Industrial*) and shipping emissions from the harbor (*V-Ni*),
presenting large contributions to $PM_{10}$. However at MSY and MSA both representatives of regional and remote backgrounds, pollutants are transported together from Barcelona urban and industrial areas resulting in an aged aerosol mixed with local pollutants.

A larger relative contribution of *Mineral* and *Aged organics* was found at the high-altitude site MSA due to a less direct exposure to anthropogenic emissions. According to previous studies (Ripoll et al., 2015; Ealo et al., 2016), a higher relative
*Mineral* contribution was found at MSA (23.6%) compared to MSY (16.2%), being relatively lower at BCN (13.6%). However, a higher absolute contribution was observed at BCN (4.6 $\mu g\ m^{-3}$), mainly from local origin. *Aged organics* also presented higher relative contribution at the MSA remote site (29.37%) compared to MSY (22.7%). However this source was not identified at BCN, where the traffic source explained the majority of the measured OC. A larger absolute and relative contribution was found for the *Aged marine* source in Barcelona (5.73; 16.9%) due to its proximity to the sea coast,



compared to MSY (1.76; 10.6%) and MSA (1.08; 11.1%). Higher contribution for A*mmonium sulfate* and *Ammonium nitrate* was found at MSY (23.7% and 7.9%) compared to MSA (9% and 7.5%), probably explained by the longer distance of the latter to the Barcelona nearby polluted area but also because of the free-troposphere conditions typically occurring in MSA during the colder period, that prevent the emplacement from the influence of the main anthropogenic emissions. The *V-Ni*

*bearing* source showed similar contributions at MSY (0.7 µg m$^{-3}$; 4.3%) and MSA (0.8 µg m$^{-3}$; 8.2%) despite the longer distance of this latter to the Mediterranean coast, pointing to a possible influence of long range transport at higher altitude affecting the mountain-top site. It should be noted that the current increasing shipping emissions are highly contributing to air quality degradation in coastal areas (Viana et al., 2014), but also in regional and remote environments as consequence of atmospheric transport processes.

A larger PM$_{10}$ source contribution mostly related with anthropogenic activities was found at BCN due to the proximity to emission sources from the metropolitan, industrial and harbor areas. Conversely, the sources with higher contribution from natural processes, such as Aged organics and Mineral, were dominant at the regional and remote backgrounds sites. Therefore, the impact of pollutants at different background sites depended on the distance to emission sources and on the transport to inland areas driven by orography and meteorology, leading to differences in the source chemical profiles

identified at the three sites. Thus, the larger the distance to emission sources the higher the aging and mixing ratio for those sources characterized by anthropogenic pollutants.

**3.2 Seasonal variation of source contribution to PM$_{10}$**

Monthly average source contributions to PM$_{10}$ obtained at the three stations are shown in Fig. 3. MSY and MSA presented a

marked PM$_{10}$ seasonal variation with higher concentrations in summer and lower in winter, in agreement with previous studies (Pérez et al., 2008 ; Ripoll et al., 2014). The summer increase is related with the higher frequency of Saharan dust events, the recirculation of air masses that prevent air renovation, the re-suspension processes owing to the dryness of soils, the low precipitation and the formation of secondary aerosols enhanced by the maximum solar radiation (Rodriguez et al., 2002). The lower winter concentrations can be explained by the high frequency of Atlantic advections leading to a higher

dispersion of pollutants and precipitation rates, but also because of the free troposphere conditions affecting the stations, predominantly at the MSA mountain-top site (Pandolfi et al., 2014).

Source contributions were proportionately distributed in BCN (10-17%), except for the *Industrial* source (3%) because most of the secondary industrial aerosols were apportioned to other secondary sources presented in this study. A notable increase of *Aged marine* (23%), *Mineral* (18%) and *V-Ni bearing* (13%) sources is given on average in summer (June, July and

August) as result of the sea breezes development, the higher frequency of Saharan dust events and the attendance of cruise ships in the Mediterranean sea during this season, respectively. By contrast, *Traffic* (23%), *Ammonium nitrate* (21%) and *Industrial* (4%) sources maximized in winter (December, January, and February) (Fig. 3a).





A larger relative contribution of secondary sources , some of them with high natural contribution, was found at the rural and remote sites (3b and 3c). *Aged organics* at MSA (29%) and *Ammonium sulfate* at MSY (24%) were the dominant sources throughout the year and reached the largest absolute contribution in summer. The *Mineral* source also presented an important contribution in summer at both sites (29% and 27%), whereas *Ammonium nitrate* (17%) at MSY and the *Industrial/Traffic*

(17%) at MSA increased during the colder period. *Ammonium nitrate* and *Ammonium sulfate* showed inverse seasonal trends, presenting this latter higher mass contribution during the warm period. These seasonal variations can be attributed to a higher SOA contribution, the favoured formation of sulfate and the nitrate gas–aerosol partitioning leading to the thermal instability of *Ammonium nitrate* during the warmer period, as was already observed in previous studies using off-line filter sampling (Pey et al., 2009; Ripoll et al., 2015) and on-line measurements (Minguillón et al., 2015; Ripoll et al., 2015).

**3.3 Scattering and absorption efficiencies of aerosol sources**

Scattering efficiencies (at 450, 525 and 635 nm) at MSY and MSA and absorption efficiencies (at 637 nm) at the three sites were obtained for the different sources identified by the PMF model (table 2). The scattering and absorption efficiencies are the regression coefficients resulting from applying a MLR model between the concentration of the sources (dependent variables) and the scattering or absorption coefficients (independent variable) (Eq. 2).

This study presents a new approach, considering the internal mixing state of atmospheric aerosols, for scattering and absorption apportionment in order to represent the real changing aerosol conditions led by atmospheric dynamics, photo-chemical reactions and physical processes. In addition, this technique allowed for the first time to calculate absorption efficiencies for all the identified aerosol sources, compared to previous studies where light absorption is entirely attributed to BC. Since no correlation was observed between source contributions pairs obtained from the PMF model (Fig. S1), we can

assume that this method also reduces some of the uncertainties regarding collinearity issues between the different variables considered in the MLR model.

The MSE and MAE of some of the sources reported in this study were not directly comparable to those coefficients published in literature for specific chemical species, given that sources identified from PMF take into account the possible aerosol internal mixing. Similar MSE were observed for *Ammonium nitrate* at MSY and MSA ($8.78\pm0.4$ and $7.84\pm0.8$ $m^2\,g^{-1}$

respectively, at 525 nm). These values were in the upper range compared to other coefficients found in the literature for the chemical compounds, highlighting the importance of taking into account the mixing state of atmospheric aerosols: Hand and Malm (2007) determined a MSE of $3.2\pm1.2$ $m^2\,g^{-1}$ for dry $PM_{2.5}$ ammonium nitrate derived by stoichiometry; Cheng et al. (2015) obtained values of $4.3\pm0.63$ $m^2\,g^{-1}$ under high mass loading in China; and Titos et al. (2013) observed a coefficient of $5\pm2$ $m^2\,g^{-1}$ for nitrate ion in an urban area in south Spain. MSE for *Ammonium sulfate* were quite different between MSY and

MSA ($4.5\pm0.2$ and $10.7\pm0.5$ $m^2\,g^{-1}$, respectively) probably due to differences in the source origin and/or to the aging and mixing of the contributions during the transport towards the stations. Hand and Malm (2007) have published lower values for the total mode of dry ammonium sulfate ranging between 0.83 and 2.4 $m^2\,g^{-1}$, whereas a coefficient of $3.5\pm0.55$ $m^2\,g^{-1}$





was found by Cheng et al. (2015) in a polluted environment. MSE for non-sea salt (nss) sulfate ion were calculated at Finokalia and Erdemli from the slope between total scattering and nss sulfate concentration, showing values of 5.9±1.8 and 5.7±1.4 $m^2 g^{-1}$ respectively (Vrekoussis et al., 2005). Higher coefficients were found in an urban background in the south of Spain (Titos et al., 2013) and for the Negev desert (Formenti et al., 2001), 7±1 and 7±2 $m^2 g^{-1}$ respectively. Given that sulfate

concentrations were mainly explained by *Ammonium sulfate* and *V-Ni*, significant differences were also observed for the MSE of the *V-Ni bearing* source at MSY and MSA (8.0±1.5 and 3.5±0.5 $m^2 g^{-1}$). *V-Ni* at MSY originated mainly from shipping emissions at regional (vessel traffic in the Mediterranean) and local (Barcelona harbor) scales. Conversely at MSA, located at higher altitude, this source may be also influenced by continental transboundary transport and then internally mixed with different species. *Aged marine* showed negative MSE at MSA at 525 and 635 nm, possibly due to the larger

distance from the sea coast and also to the $PM_{2.5}$ cut-off inlet which prevents the sampling of coarse particles. However, coefficients at MSY (1.21±0.32 $m^2 g^{-1}$) showed values within the same range than those reported by Hand and Malm (2007) for coarse mode sea salt (1.0 $m^2 g^{-1}$). MSE for *Aged organics* (1.4±0.2 and 1.3±0.3 $m^2 g^{-1}$) and *Mineral* sources (1.3±0.1 and 1.1±0.1 $m^2 g^{-1}$) at MSY and MSA, respectively, showed large similarity between both sites. This fact can be explained by the common stability of this kind of particles during the atmospheric transport, but also as a result of the similar source origin

governed by same atmospheric processes. The *Mineral* source is mainly attributed to the transport of Saharan dust particles reaching the stations and local/regional dust re-suspension processes. Lower coefficients for mineral matter were shown by Hand and Malm (2007) (0.7±0.2 $m^2 g^{-1}$), by Titos et al. (2013) in Granada urban background (0.2±0.3 $m^2 g^{-1}$) and by Vrekoussis et al. (2005) in Erdemli (0.2 $m^2 g^{-1}$). Similar coefficients were obtained by Vrekuossis et al. (2005) for Finokalia (1 $m^2 g^{-1}$), and by Pereira et al. (2008) and Wagner et al. (2009) for mineral dust in Portugal, 1±0.1 $m^2 g^{-1}$ and 0.9 $m^2 g^{-1}$

respectively. On the other hand, the *Aged organics* source originates from a mixture of local biogenic emissions and the transport of pollutants from the Barcelona area, giving rise to an aged and mixed aerosol from regional/local origin, showing a low scattering efficiency as a result of the EC contained within in the source. A similar coefficient (1.4 $m^2 g^{-1}$) for the total mode of primary organic matter (POM) was reported by Hand and Malm (2007) and a higher coefficient (4.5±0.73 $m^2 g^{-1}$) was found by Cheng et al. (2015) during a pollution episode in China. The *Industrial/Traffic* source showed similar

coefficients at MSY (2.1±0.8 $m^2 g^{-1}$) and MSA (2.3±0.5 $m^2 g^{-1}$) likely due to a common source origin affecting both sites from the Barcelona urban and industrial areas, where pollutants are transported to inland regions undergoing similar transformation processes. It is remarkable that MSE for some of the sources identified in this work which highly contribute to air quality degradation, such as *Industrial/Traffic* or *V-Ni*, are not available in the literature.

Interestingly, a higher scattering wavelength dependence was observed for those sources with higher contribution from

anthropogenic tracers which are mainly present in the fine mode (Table 2). The Scattering Ångström Exponent (SAE) was obtained as a linear fit of 3λ scattering in the 450–635 nm range. This parameter provide information on the size of the particles; generally a SAE lower than 1 or higher than 2 indicates that the scattering is dominated by larger or finer particles, respectively (Schuster et al., 2006). *Aged organics* and *V-Ni* sources showed the highest SAE at MSY (2.2 and 2.4) and MSA (3.6 and 2.2), respectively, pointing to a predominance of fine particles within these sources. Previous studies have





demonstrated the strong contribution from shipping emissions to fine aerosols (Viana et al., 2009), and especially to ultrafine particles (Saxe and Larsen, 2004). Conversely, the lowest SAE was found for *Mineral* and *Aged marine*, which are mainly contained within the coarse fraction.

Prior studies focusing on absorption efficiencies are mainly referred to BC particles and to the possible coating with non-absorbing material. To the author's knowledge, this is the first time that absorption efficiencies were calculated for different aerosol sources, aiming to study the relationship between the main contributors to air quality degradation and their potential to absorb the light. MAE coefficients at 637 nm for the three sites are summarized in Table 2. The highest absorption efficiencies were found for the *Traffic* source identified at BCN ($1.672\pm0.050$ m$^2$ g$^{-1}$) and the equivalent *Industrial*/*Traffic* at MSY ($0.867\pm0.047$ m$^2$ g$^{-}$1) and MSA ($0.206\pm0.02$ m$^2$ g$^{-1}$), due to the internal mixing with BC particles from fossil fuel combustion. Interestingly, the *V-Ni bearing* source which highly contributes to light scattering also exhibited a high MAE at BCN ($0.928\pm0.058$ m$^2$ g$^{-1}$) with decreasing coefficients at MSY ($0.526\pm0.065$ m$^2$ g$^{-1}$) and MSA ($0.165 \pm 0.017$ m$^2$ g$^{-1}$). This source is progressively becoming more relevant for air quality due to the increased shipping emissions in recent years (Viana et al., 2014), but also leads to an important effect on light absorption as consequence of the internal mixing with combustion aerosols. A large MAE was observed for *Ammonium nitrate* at MSA ($0.364\pm0.023$ m$^2$ g$^{-1}$) which explained around 20% of the measured EC concentration (Fig. 2a), compared to BCN and MSY ($0.28\pm0.040$ and $0.234\pm0.028$ m$^2$ g$^{-1}$) where this source presents a local and regional origin. Recently, Ripoll et al. (2015) have shown the increased concentration of nitrate, ammonium, EC and traffic/industrial tracers under European scenarios, when polluted air masses from central and eastern Europe are transported at higher altitude to the MSA site. This fact may explain the higher absorption efficiency found for *Ammonium nitrate* at this site. Lower MAE were found for *Ammonium sulfate* ($0.359\pm0.035$, $0.122\pm0.010$ and $0.173\pm0.021$ m$^2$g$^{-1}$) at BCN, MSY and MSA. Higher absorption efficiencies were observed for the main anthropogenic sources at BCN, where fresh primary pollutants, mostly composed by darker particles, are emitted within the metropolitan, industrial and harbor areas. However, lower MAE were determined for the same pollutant sources at MSY and MSA pointing to a decrease on absorption efficiency towards inland areas as consequence of the mixing and aging of pollutants during the transport towards the stations. Aerosol sources dominated by natural contributions showed the lowest MAE at the three sites. The *Road re-suspension* source found at BCN, which is partially composed by mineral matter, exhibited low MAE ($0.062 \pm0.084$ m$^2$ g$^{-1}$). The *Mineral* source presented the lowest MAE at BCN and MSY ($0.09\pm0.05$ and $0.005\pm0.007$ m$^2$ g$^{-1}$), showing also low coefficients at MSA ($0.03\pm0.003$ m$^2$ g$^{-1}$). Coefficients in the same order of magnitude at 660 nm were found for the Sahara-Sahel and Gobbi deserts ranging between 0.01 and 0.02 m$^2$ g$^{-1}$ (Alfaro et al., 2004) and for El Cairo and Morocco, $0.02\pm0.004$ and $0.06\pm0.014$ m$^2$ g$^{-1}$ respectively (Linke et al., 2006). *Aged marine* also exhibited low absorption coefficients at BCN, MSY and MSA ($0.108\pm0.021$, $0.027\pm0.018$ and $0.015\pm0.010$ m$^2$ g$^{-1}$), being larger at BCN due to a possible mixing with darker particles at urban level. *Aged organics* showed similar MAE at MSY and MSA ($0.169\pm0.011$, $0.140\pm0.009$ m$^2$ g$^{-1}$, respectively) due to the local/regional origin of this source with similar composition at both sites. The absorption efficiency of this latter source is mainly explained by the EC contained within the source chemical profile, but also might be partially due to the presence of light absorbing material detected as OC, such as brown carbon (Putaud et al., 2014).



### 3.4 Seasonal variation of source contributions to scattering and absorption

Monthly source contributions to the total scattering and absorption coefficients are shown in Fig.3. The partial scattering and absorption apportioned to each source was calculated as the product between the corresponding MSE or MAE and the

aerosol source concentration (Eq. 1 and 2). Accordingly to the scattering efficiencies previously reported in Table 2, average scattering for the whole period appears to be mainly dominated by *Ammonium sulfate* (36% and 35%) and *Ammonium nitrate* (24% and 21%) at MSY (Fig. 3g) and MSA (Fig. 3h) respectively. Both sources presented inverse seasonal cycles following the seasonal variation of mass contributions, with *Ammonium sulfate* maximizing in summer at MSY (46%) whereas showing similar contribution throughout the year at MSA. Conversely, *Ammonium nitrate* mainly governed the light

scattering in winter (42% and 29% at MSY and MSA). *V-Ni* also exhibited substantial contribution to scattering in summer (16%) despite the relative low mass contribution (5% and 10% at MSY and MSA). Less relevant were the scattering contributions of the others sources.

Light absorption appeared to be almost dominated by the *Traffic* source at BCN and in a minor proportion by the equivalent *Industrial/Traffic* at MSY and MSA (Fig. 3d, e, f), showing high contributions in winter (65%, 42%, 22%) despite the

relative low mass concentration (23%, 11%, 17%). Interestingly, the *V-Ni bearing* source also played an important role in the light absorption reaching the maxima in summer, due to the increased cruise ships emissions in the Mediterranean but also because of the more intensive sea breezes circulation; average contributions during this season were 31% at BCN and around 17% at MSY and MSA. Therefore, *Traffic*, *Industrial/Traffic* and *V-Ni* sources which highly influence air quality also have caused an important effect on radiative forcing, particularly in those sites closer to the emission sources. *Aged organics*

became a relevant source in the absorption process at the regional and continental background sites (20% and 32%) due to its large contribution in PM$_{10}$. *Ammonium sulfate* contributed on average by 10%, 16% and 12% at BCN, MSY and MSA, whereas *Ammonium nitrate* showed an increasing absorption towards inland areas (8%, 10% and 21%, respectively) markedly maximizing in winter.

As a summary, we have shown that the main target pollutant sources affecting air quality degradation have caused important

effects on light extinction in the northwestern Mediterranean. Increasing absorption efficiencies were found for the main anthropogenic factors in areas closer to the emission sources, pointing that the aging and mixing state of aerosols resulting from the atmospheric transport and transformation processes are key factors influencing light absorption. Light absorption was mainly governed by the *Traffic* source at BCN followed by *V-Ni*. These sources at MSY and MSA were relevant in the light absorption process but not dominant, probably due to the mixing and aging of combustion pollutants during the

transport towards the stations. *Ammonium sulfate*, *Ammonium nitrate* and *Aged organics* however, gained relative importance in the absorption process at MSY and MSA. Accordingly to the reviewed studies, light scattering appeared to be dominated by *Ammonium sulfate* and *Ammonium nitrate*, in summer and winter respectively. As a novelty, *V-Ni* bearing



source (mainly originated from shipping emissions) was also found to be an important contributor to scattering during summer, due to the internal mixing with sulfate particles.

### 3.5 Reconstruction of scattering, absorption and SSA time series

Scattering and absorption time series were reconstructed by means of the sum of the partial scattering or absorption contributions determined for each source. In this study, the partial scattering and absorption coefficients were obtained from the product between the daily source concentration and the specific MSE/MAE computed by means of the MLR (Eq. 1 and 2).

Strong correlations were found between the measured and modeled extensive optical parameters for the period 2010-2014 at

BCN and MSY, and for the period 2011-2014 at MSA, when optical measurements were available (Fig. 4). Results showed good agreement for scattering at 525 nm at MSY ($R^2$=0.88) and MSA ($R^2$=0.92). Absorption at 637 nm also exhibited good correlation by comparing measured and predicted coefficients at BCN ($R^2$=0.81), MSY ($R^2$=0.80) and MSA ($R^2$=0.93). Slopes were close to one in all the cases and ranged between 0.96 and 0.98. These results are consistent with the good agreement obtained in the MLR model for MSE and MAE calculation, showing a $R^2$ of 0.96 in all the cases and ensuring the

robustness of the regression coefficients computed for each site. The fractional bias (FB) between the measured and predicted coefficients for each sampling site was obtained following equation 3 and results are shown in Fig. 5, where the FB is broken down by quintile from lowest to highest scattering and absorption values (Ryan et al., 2005). According to published results, a consistent overestimation was observed for all the modeled coefficients in the lower range of scattering and absorption, showing the highest bias in the 1st quintile. Biases were substantially reduced in the median range values,

whereas a minor underestimation was observed for the highest scattering and absorption values, 4th and 5th quintiles, with a negative FB. On average, a 3.8% and 5.6% overprediction was obtained for the modeled absorption coefficients at MSY and BCN, using 503 and 375 daily data points in the analysis. Scattering overprediction at MSY was 4% using 307 daily points, whereas at MSA scattering and absorption coefficients biased by 30.9% and 19.9% the observed values considering 220 and 303 data points in the analysis. A larger overestimation of the measured coefficients at MSA might be explained mainly by

the less number of daily chemical data used in the PMF model for the quantification of source contributions, but also because of the less scattering and absorption data points available for the MLR analysis. A consistent improvement in the reconstruction of scattering and absorption coefficients was achieved using the PMF-MLR technique at BCN and MSY, thus taking into account the possible internal mixing of atmospheric particles, and being the number of data points used in the PMF and MLR models limiting factors for a solid performance of the analysis. As a result, long-term time series of

scattering and absorption were satisfactory reconstructed when chemical data was available, for the period 2004-2014 at BCN and MSY and for the period 2011-2014 at MSA (Fig. 6).

Additionally, the simulation of the SSA at MSY was obtained from the ratio between the modeled scattering (at 635 nm) and extinction coefficients (Fig. 7). In order to prevent biased results due to the low signal of the instruments during cleaner





atmospheric conditions, only values higher that 1 and 0.1 $Mm^{-1}$ for scattering and absorption coefficients, respectively, were considered in the SSA calculation. A higher uncertainty was found for the simulation of SSA compared to the extensive parameters (Fig. 4) with a $R^2=0.43$; nevertheless slope was close to one. SSA values obtained for each source are summarized in Table 2 and provide information on the relative importance of scattering or absorption in the light extinction

process. As expected, the sources internally mixed with combustion particles from anthropogenic activities such as *Industrial/Traffic*, *Aged organics* and *V-Ni* exhibited lower SSA, 0.74, 0.84 and 0.9 respectively. Conversely, *Aged marine* and *Mineral* sources showed the highest coefficients, 1 and 0.98 respectively, leading to a scattering dominance in the light extinction process. Accordingly to studies in the literature the *Mineral* source showed a SSA close to 1. Linke et al. (2006) showed values around 0.98-0.99 at 532 nm and lower coefficients were found by Müller et al. (2011) for mineral dust (0.96)

and marine (0.95) aerosols at 530 nm. Note that equivalent wavelengths should be considered when comparing SSA mineral dust with coefficients in literature due to the strong wavelength dependence.

### 3.6 Long-term trends

The good agreement found between modeled and measured optical parameters allowed for the reconstruction of the

scattering and absorption coefficients since 2004 at BCN and MSY and since 2010 at MSA (Fig. 6), when chemical speciation data was available. As a result, long-term trends of optical parameters and their relationship with trends of aerosol sources can be investigated for an 11-year period at MSY. Despite a larger uncertainty was found for the modeled SSA, this technique allowed to further investigate the temporal trend of this important parameter and its relation with changes in atmospheric composition (Fig. 7). Temporal trends at BCN and MSA were not evaluated due to a change in the location of

the BCN sampling station, which was firstly installed close to a parking area until 2009 and then moved to an urban background (Pandolfi et al., 2016). The short time series available of chemical composition at MSA makes unfeasible the simulation of optical parameters for at least a 10-year period, and consequently also the trend study.

Temporal trends were analyzed by means of the Theil-Sen (TS) slope estimates for the modeled deseasonalized monthly averages of optical parameters at MSY during the period 2004-2014 (Table3). The monthly time variation of absorption,

scattering and SSA were decomposed in trend, main component, seasonal component and residuals using the multi-exponential (ME) method (Figure 8). Linear trends were identified for all the studied variables, thus the non-linearly (NL) parameter was less than 10% (Shalatov et al., 2015). Statistically significant decreasing trends were found in all the cases. Absorption at 637 nm has been reduced by -4.1 % $y^{-1}$ (-0.16 $Mm^{-1}$ $y^{-1}$). A decrease of -4.6 % $y^{-1}$ (-2.14 $Mm^{-1}$ $y^{-1}$) was obtained for scattering, and very similar trends were observed for 450 and 635 nm. Accordingly to this results, decreasing

trends were also observed for all the aerosol sources identified at MSY during 2004-2014, except for *Aged organics* and *Aged marine* (Pandolfi et al., 2016). A reduction in the absorption coefficient is directly related with the significant decreasing trends found by Pandolfi et al. (2016) for strong light-absorbing sources such as the *Industrial/Traffic* (-0.1 µg m$^{-3}$ $y^{-1}$) and *V-Ni bearing* (-0.07 µg m$^{-3}$ $y^{-1}$). However, scattering decreasing trends were probably associated with a reduction



in those sources which scattered light more efficiently, *Ammonium sulfate* and *Ammonium nitrate*. In Pandolfi et al. (2016) these sources decreased by -0.32 and -0.15 µg m$^{-3}$ y$^{-1}$, respectively, showing the highest reduction at MSY. A marked decline was also observed for nitrate and sulfate PM in other European monitoring sites since 1990, as outlined in the EMEP report 1/2016.

Further explanation on the causes leading to a reduction in most of the sources identified at MSY is detailed in Pandolfi et al. (2016). A decrease in *Ammonium nitrate* can be explained by the reduction of ambient NO$_X$ concentrations, as consequence of the lower energy consumption related with the financial crisis and the associated decrease in NO$_X$ emissions from the five power generation plants surrounding Barcelona. A reduction in NO$_X$ concentrations is also related to a slight recession of the traffic density and therefore on road traffic emissions, as consequence of the financial crisis. A reduction in *Ammonium*

*sulfate* resulted mainly from the decrease of secondary sulfate, which is mainly attributed to the EC Directive on Large Combustion Plants implemented in Spain from 2008, resulting in the flue gas desulfurization (FGD) at several facilities. A particular case is presented for *V-Ni*; this source is internally mixed with secondary sulfate and combustion aerosols and consequently contributed simultaneously to both light scattering and absorption. A reduction in *V-Ni* can be explained by the implementation of the 2008 regional air quality plan, in which the use of heavy oils and petroleum coke for power generation

was forbidden around Barcelona in favour of natural gas. However, the enlarged and uncontrolled shipping emissions taking place in the last years might lead to an increase of the relative weight of shipping to the total of anthropogenic emissions (Viana et al., 2014) and therefore to the *V-Ni* bearing source. Hence, despite the observed reduction in *V-Ni*, the increasing shipping emissions might cause important changes in the internal mixture of this relevant source leading to changes in light extinction and thus becoming a key pollutant source to consider for climate assessment.

Interestingly, the SSA showed a significant decreasing trend of -0.11 % y$^{-1}$ (-0.001 yr$^{-1}$) leading to a total reduction (TR) of -1.24 % since 2004 at MSY, pointing that the atmosphere is getting significantly darker. A more significant decreasing trend on SSA was found by Putaud et al. (2014) from ground base measurements at dry conditions at 700 nm (-0.7 % y$^{-1}$) and from columnar measurements using sun-photometer retrievals at 675 nm (-0.8 % y$^{-1}$) in the Po Valley. Differences in the SSA reduction at both sites might be explained by the severe pollution episodes taking place in the Po Valley resulting in a higher

dominance of absorption in the light extinction process, compared to MSY which is representative of a less polluted environment.

Several facts might have accounted for a decreasing trend in the SSA at MSY since 2004; the air quality policies implemented in the last years were very effective in reducing the atmospheric concentration of aerosol species with large scattering efficiency (cooling effect), leading to a scattering TR of -52% since 2004. However, abatement strategies focused

on reducing pollutants containing dark particles (mainly emitted from fossil fuel combustion sources) were less effective, leading to a less pronounced TR of -44% in the absorption coefficient. This fact resulted in the enhancement of the relative weight of absorption, therefore contributing to diminish the SSA. It should also be considered that the reduction of the primary energy consumption and industrial activities related to the financial crisis affecting Spain since 2008 contributed to the decrease of the ambient concentration of pollutants (Querol et al., 2014), and consequently to the reduction of light



scattering and absorption. Another fact to be considered is the increasing use of commercial and domestic biomass burning system in most European countries. Putaud et al. (2014) found that an increase in the contribution of light-absorbing organic matter, from biomass burning emissions, to light absorption during cold months could be an explanation for the decrease in SSA in the Po Valley. A reduction in the SSA at MSY might be partially explained by a more significant decrease in the

sources scattering light more efficiently, which could have helped to unmask the contribution of biomass burning to light absorption. Minguillón et al. (2015) and Ealo et al. (2016) found that winter biomass burning emissions at MSY present an important contribution to both ambient air BC and organic aerosols (OA) concentrations. This fact, together with the observed increasing trend for *Aged organics* (Pandolfi et al., 2016), might partially explain a less pronounced decreasing trend in the absorption coefficient contributing to reduce the SSA.

The reduction in the SSA trend points out the increasing prominence of absorption in the light extinction process in the NW Mediterranean. The decrease in the atmospheric PM load was thought to be beneficial for air quality and health; however the resulting reduction of light-scattering aerosols lead to an increase of the incoming solar radiation contributing to climate warming. This radiative forcing is enhanced by the less effectiveness of the air quality policies focused on reducing pollutants containing BC particles, which highly contribute to light absorption. Therefore, the heterogeneous and non-

selective reduction of pollutant sources, showing opposite effects on radiative forcing, may cause important changes on the temporal trends of optical properties and therefore on the earth-atmosphere radiative balance. This fact highlights the importance of implementing win-win regulation strategies which should be focused on both air quality and climate, and not acting separately. Given the high $PM_{10}$ concentration, the toxicity of their chemical tracers and the contribution to light absorption in the NW Mediterranean, *Industrial/Traffic* and *V-Ni bearing* are target sources which should be abated for

preventing health disorders, improving air quality and mitigate climate warming.

### 4. Summary and conclusions

Mass scattering and absorption efficiencies (MSE and MAE) of different aerosol sources were investigated at urban, regional and remote backgrounds in the NW Mediterranean using unique large datasets of $PM_{10}$ chemical speciation and optical

properties. With this aim, a new approach was developed for scattering and absorption source apportionment considering the aerosol source contributions, arising from a positive matrix factorization (PMF) analysis, as dependent variables in a Multilinear Regression (MLR) model for MSE and MAE calculation.

Seven sources were identified at the Montsec (MSA) remote mountain-top site, where *Aged organics* (29.4%) was the foremost constituent of $PM_{10}$, followed by, *Mineral* (23.6%), *Industrial/Traffic* (11.3%) *Aged marine* (11.1%), *Ammonium*

*sulfate* (9%), *V-Ni bearing* (8.2%) and *Ammonium nitrate* (7.5%). Same sources have been identified at Montseny (MSY), showing the secondary aerosol sources a higher relative load at these background sites. At the Barcelona (BCN) urban background site *Aged organics* was not identified, however specific pollutant sources related to the direct anthropogenic emissions (*Traffic*, *Industrial/metallurgy* and *Road-resuspension*) were isolated. Some of the anthropogenic sources




identified at MSY and MSA, such as *Industrial/Traffic*, resulted from the internal mixture between local aerosols and pollutants transported from the BCN industrial and metropolitan area. We have found that the impact of aerosol sources and the changing chemical profile at different background sites depended on the distance and transport of pollutants to inland areas driven by orography and meteorology.

The highest absorption efficiencies were attributed to sources internally mixed with BC particles. *Traffic* ($1.7 \ m^2 \ g^{-1}$) at BCN or the equivalent *Industrial/Traffic* at MSY and MSA (0.87 and $0.2 \ m^2 \ g^{-1}$) mainly governed light absorption in the NW Mediterranean, with increasing contributions in winter by 54%, 40% and 18%, respectively. The *V-Ni bearing* source was the second most efficient light-absorbing source in BCN ($0.93 \ m^2 \ g^{-1}$), showing also a notable efficiency at MSY and MSA (0.53 and $0.16 \ m^2 \ g^{-1}$). This source originated mainly from shipping emissions and highly contributed to both scattering

(around 16% at the background sites) and absorption (31%, 17% and 16%, respectively) during summer, due to the simultaneous internal mixing with sulfate and combustion aerosols. These sources were relevant but not dominant at MSY and MSA, where secondary aerosol sources (*Ammonium sulfate*, *Ammonium nitrate* and *Aged organics*) gained relative importance in the light extinction process. *Aged organics* contributed on average to absorption by 20% and 32% at MSY and MSA. *Ammonium nitrate* showed increasing contributions in winter, accounting for 11%, 17% and 21% of the total

absorption at BCN, MSY and MSA. A high spatial variability of MAE was observed for most of the anthropogenic sources, from high values at the BCN site to decreasing coefficients at the background stations, pointing that the aging and mixing state of aerosols are key factors influencing light absorption. The highest scattering efficiencies were found for *Ammonium sulfate* (4.5 and $10.7 \ m^2g^{-1}$), *Ammonium nitrate* (8.8 and $7.8 \ m^2 \ g^{-1}$) and *V-Ni* (8 and $3.5 \ m^2 \ g^{-1}$) at MSY and MSA respectively, dominating the scattering throughout the year with marked seasonal trends. *Ammonium nitrate* highly

contributed in winter by 42% and 29%, respectively, whereas in summer scattering was mainly governed by *Ammonium sulfate* (46% and 35%).

Sources internally mixed with relatively dark and fine particles and highly contributing to light absorption, such as *Industrial/Traffic, Aged organics* and *V-Ni*, were simultaneously characterized with low single scattering albedo (SSA) and high scattering Ångström exponent (SAE). Conversely, *Mineral* and *Aged marine* showed the highest SAE and the lowest

SSA, being scattering the dominant process in the light extinction. These findings for the intensive parameters were consistent at MSY and MSA. The variability of these intensive optical properties as a function of both chemical composition and emissions sources is a valuable information that could help to constrain model estimates of aerosol parameters.

Significant decreasing trends were observed for the modeled scattering ($-4.6 \ \% \ y^{-1}$), absorption ($-4.1 \ \% \ y^{-1}$) and SSA ($-0.11 \ \% \ y^{-1}$) time series during 2004-2014 at MSY. A total reduction (TR) of -1.12% in the SSA was mainly motivated by the

heterogeneous and non-selective reduction of key aerosol sources showing opposite effects on radiative forcing. Abatement strategies aimed at reducing atmospheric PM were thought to be beneficial for air quality and health. However, these measures have resulted in a more pronounced reduction of light-scattering aerosol sources (*Ammonium nitrate*, *Ammonium sulfate*), leading to an increase of the incoming solar radiation and therefore contributing to climate warming. This positive radiative effect is enhanced by the less effectiveness of air quality strategies for reducing light-absorbing sources containing





dark particles. A decrease in the SSA trend points to a darkening of the atmosphere and consequently to a progressive predominance of absorption in the light extinction process in the NW Mediterranean. Accordingly to the results presented in this work, future strategies need to focus on preferentially reducing atmospheric aerosols mainly originated from combustion sources. *Industrial/Traffic* and *V-Ni* aerosol sources, which highly contributed to air quality degradation but also to light absorption, should be abated thus addressing win-win policies aimed to improve air quality and mitigate climate warming in the NW Mediterranean.

The author's would like to highlight the new findings and main advantages of the PMF-MLR technique, summarized as follows. This study can be reproduced in other emplacements with available large chemical data sets in order to further research the role of aerosol sources on light extinction, contributing to reduce uncertainties in climate forcing.

- Using aerosol source contributions in the MLR model allows for a comprehensive performance of the chemical speciation in the apportionment to scattering and absorption, taking into account the internal mixing state of atmospheric aerosols and therefore achieving a more realistic proxy for MSE and MAE coefficients.
- Scattering and absorption efficiencies for target pollutant sources contributing to air quality degradation also revealed a substantial contribution to light extinction in the NW Mediterranean, thus leading to directly linking the air quality and climate effects of airborne particle matter. To the author's knowledge, this is the first time that light absorption is apportioned to different emission sources, compared to previous studies where it is entirely attributed to BC particles.
- A consistent improvement in the reconstruction of scattering coefficients was achieved regarding previous studies. Correlation coefficients were higher than 0.8 with slopes close to 1.0 between modeled and measured scattering and absorption. The number of daily data used in the PMF and MLR models was found to be a limiting factor in the performance of the analysis, resulting in under and overestimation for the low and high scattering and absorption coefficient values, respectively.
- Scattering, absorption and single scattering albedo (SSA) time series were reconstructed since 2004 based on the $PM_{10}$ source contribution data and MSE and MAE regression coefficients. As a result, a long term trend study was performed for these climate relevant parameters allowing to further research on the effects that atmospheric pollutants and the abatement strategies implemented in the last years are causing on atmospheric composition and light extinction.




## Acknowledgements

This work was supported by the MINECO (Spanish Ministry of Economy and Competitiveness) and FEDER funds under the PRISMA project (CGL2012-39623-C02/00), by the MAGRAMA (Spanish Ministry of Agriculture, Food and Environment) and by the Generalitat de Catalunya (AGAUR 2014 SGR33 and the DGQA). This work has received funding from the European Union's Horizon 2020 research and innovation programme under grant agreement No 654109. Marco Pandolfi is funded by a Ramón y Cajal Fellowship (RYC-2013-14036) awarded by the MINECO. The authors would like to express their gratitude to D. C. Carslaw and K. Ropkins for providing the OpenAir software used in this paper (Carslaw and Ropkins, 2012; Carslaw, 2012).

## Data availability

The Montseny and Montsec data sets used for this publication are accessible online on the WDCA (World Data Centre for Aerosols) web page: http://ebas.nilu.no. The Barcelona data sets were collected within different national and regional projects and/or agreements and are available upon request.

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





**Figure captions**

**Figure 1.** Map location and topographic profiles of Barcelona (BCN; urban background), Montseny (MSY; regional background) and Montsec (MSA; remote mountain-top background) measurement sites.

**Figure 2. (a)** Source chemical profiles and **(b)** source contributions to PM$_{10}$ mass concentration obtained at MSA by means of the PMF model. PM$_{10}$ average concentration, and absolute (µg m$^{-3}$) and relative (%) source contributions are reported for the study period (2010-2014).

**Figure 3.** Relative (%) monthly average source contribution to: PM$_{10}$ concentration (µg m$^{-3}$) (a, b and c), absorption (Mm$^{-1}$) at 637 nm (d, e and f) and scattering (Mm$^{-1}$) at 525 nm (g and h). PM$_{10}$ source contributions were obtained from the PMF model, whereas scattering and absorption contributions have been modelled by means of the PMF-MLR technique. The study period ranges between 2004-2014 at BCN and MSY and between 2010-2014 at MSA.

**Figure 4.** Relationship between measured and modeled optical parameters: absorption at 637 nm (a, b, and c for MSA, MSY and BCN respectively), scattering at 525nm (d and e for MSA and MSY) and SSA at MSY (f).

**Figure 5.** Average fractional bias (FB) calculated for the modeled and measured scattering (Sc) and absorption (Abs) coefficients at BCN, MSY and MSA broken down by quintile from the lowest to highest scattering and absorption coefficient values. "n" accounts for the number of daily data points used in the FB calculation.

**Figure 6.** Time series of the daily average modeled and measured extensive optical coefficients (scattering at 525 nm and absorption at 637 nm) for (a) BCN and (b) MSY during the period 2004-2014, and for (c) MSA during the period 2010-2014.

**Figure 7.** Boxplots time series of the modeled (white box) and measured (grey box) annual average of SSA at MSY during the period 2004-2014.

**Figure 8.** Temporal trends of the monthly average of absorption at 637 nm, scattering at 525 nm and SSA obtained by means of the multi-exponential test at MSY during the period 2004-2014. The time series were decomposed in: simulated coefficient (green), trend (red), main component (black), seasonal component (blue) and residue (grey).




**Table 1.** Absolute (µg m$^{-3}$) and relative (%) average source contribution to PM$_{10}$ at BCN and MSY during the period 2004-2014 (Pandolfi et al., 2016), and at MSA during the period 2010-2014.

| (µg m$^{-3}$; %) | PM$_{10}$ | Aged marine | Mineral | Aged organics | Amm. nitrate | Amm. sulfate | Industrial/ Traffic | Industrial/ Metallurgy | V-Ni | Traffic | Road resuspension |
|---|---|---|---|---|---|---|---|---|---|---|---|
| BCN | 34.0; 100 | 5.73; 16.9 | 4.61; 13.6 | | 4.45; 13.1 | 4.67; 13.7 | | 0.96; 2.8 | 3.32; 9.8 | 5.14; 15.1 | 4.25; 12.5 |
| MSY | 16.7; 100 | 1.76; 10.6 | 2.70; 16.2 | 3.78; 22.7 | 1.31; 7.9 | 3.95; 23.7 | 1.43; 8.6 | | 0.71; 4.3 | | |
| MSA | 9.65; 100 | 1.08; 11.1 | 2.27; 23.6 | 2.84; 29.4 | 0.72; 7.5 | 0.87; 9.0 | 1.09; 11.3 | | 0.79; 8.2 | | |





**Table 2.** Scattering and absorption efficiencies (MSE and MAE; $m^2\,g^{-1}$) calculated for the different aerosol sources identified by PMF at BCN, MSY and MSA in the $PM_{10}$ fraction. Scattering Ångström exponent (SAE) and single scattering albedo (SSA) coefficients were obtained for each source at MSY and MSA. Note that SAE was not considered for the *Aged marine* source at Montsec due to the $PM_{2.5}$ cut-off inlet. The study period ranges between 2010-2014 at BCN and MSY and 2011-2014 at MSA.

*SAE for the *Industrial/Traffic* source at MSY was calculated in the range 450-525 nm.

| | | Aged marine | Mineral | Aged organics | Amm. nitrate | Amm. sulfate | Industrial/ Traffic | Industrial/ Metallurgy | V-Ni | Traffic | Road resuspension |
|---|---|---|---|---|---|---|---|---|---|---|---|
| BCN | MAE 637 | 0.108 ± 0.021 | 0.087 ± 0.050 | | 0.284 ± 0.040 | 0.359 ± 0.035 | | 0.138 ± 0.185 | 0.928 ± 0.058 | 1.672 ± 0.050 | 0.062 ±0.084 |
| MSY | MSE 450 | 1.205 ± 0.385 | 1.046 ± 0.130 | 1.990 ± 0.258 | 10.456 ± 0.494 | 5.860 ± 0.256 | 2.241 ± 0.982 | | 10.844 ± 1.850 | | |
| | MSE 525 | 1.211 ± 0.316 | 1.262 ± 0.106 | 1.414 ± 0.212 | 8.783 ± 0.405 | 4.508 ± 0.210 | 2.057 ± 0.805 | | 8.029 ± 1.516 | | |
| | MSE 635 | 1.201 ± 0.284 | 1.429 ± 0.096 | 0.916 ± 0.190 | 6.980 ± 0.364 | 3.092 ± 0.188 | 2.425 ± 0.723 | | 4.687 ± 1.362 | | |
| | MAE 637 | 0.027 ± 0.018 | 0.005 ± 0.007 | 0.169 ± 0.011 | 0.234 ± 0.028 | 0.122 ± 0.010 | 0.867 ± 0.047 | | 0.526 ± 0.065 | | |
| | SAE | 0.010 | -0.896 | 2.254 | 1.175 | 1.861 | 0.556* | | 2.451 | | |
| | SSA | 0.978 | 0.997 | 0.844 | 0.968 | 0.962 | 0.736 | | 0.899 | | |
| MSA | MSE 450 | 0.036 ± 0.407 | 0.931 ± 0.115 | 2.114 ± 0.338 | 9.839 ± 0.978 | 13.825 ± 0.792 | 2.714 ± 0.644 | | 4.823 ± 0.659 | | |
| | MSE 525 | (-)0.054 ± 0.332 | 1.077 ± 0.093 | 1.335 ± 0.275 | 7.839 ± 0.797 | 10.699 ± 0.537 | 2.354 ± 0.525 | | 3.538 ± 0.537 | | |
| | MSE 635 | (-)0.036 ± 0.268 | 1.276 ± 0.076 | 0.617 ± 0.223 | 6.006 ± 0.645 | 7.439 ± 0.522 | 2.044 ± 0.425 | | 2.274 ± 0.435 | | |
| | MAE 637 | 0.015 ± 0.010 | 0.029 ± 0.003 | 0.14 ± 0.009 | 0.364 ± 0.023 | 0.173 ± 0.021 | 0.206 ± 0.016 | | 0.165 ± 0.017 | | |
| | SAE | - | -0.914 | 3.594 | 1.432 | 1.804 | 0.819 | | 2.189 | | |





**Table 3.** Theil-Sen (TS) trends at a 95% confidence level for deseasonalized monthly averages time series of scattering, absorption and single scattering albedo (SSA) coefficients at MSY during the period 2004-2014. AR (Mm$^{-1}$; %)= Average reduction; TR (%)= Total reduction; The significance of the trends (p-value trend) was obtained by means of TS method using monthly averages: *** (p-value < 0.001), ** (p-value < 0.01), * (p-value <0.05).
The Non-linearity parameter (%) was calculated by means of the multi-exponential (ME) test.

| | | TS | | | | ME |
|---|---|---|---|---|---|---|
| | | p-value slope | AR (Mm$^{-1}$ yr$^{-1}$) | AR (% yr$^{-1}$) | TR (%) | NL (%) |
| MSY | Sc 525 | *** | -2.14 | -4.57 | -52 | 5.56 |
| | Abs 637 | *** | -0.16 | -4.13 | -44 | 4.17 |
| | SSA | *** | -0.001 | -0.11 | -1.24 | 0.002 |



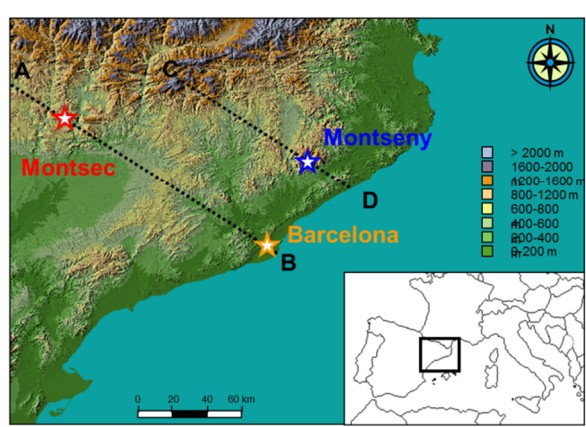

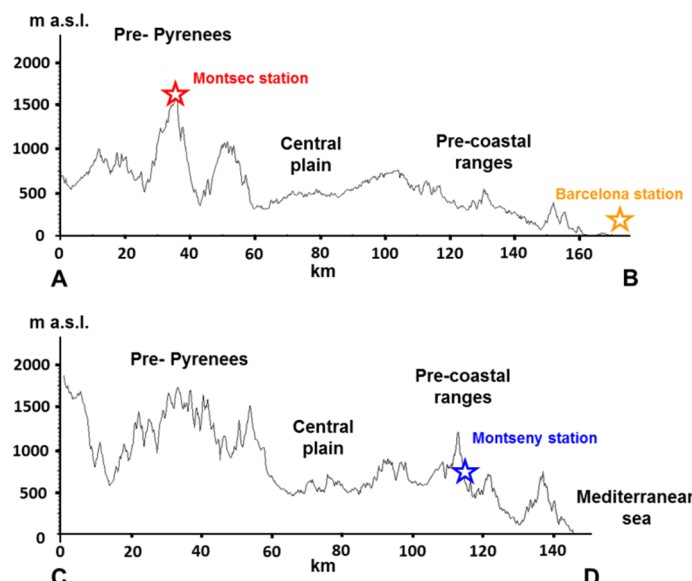

**Figure 1**



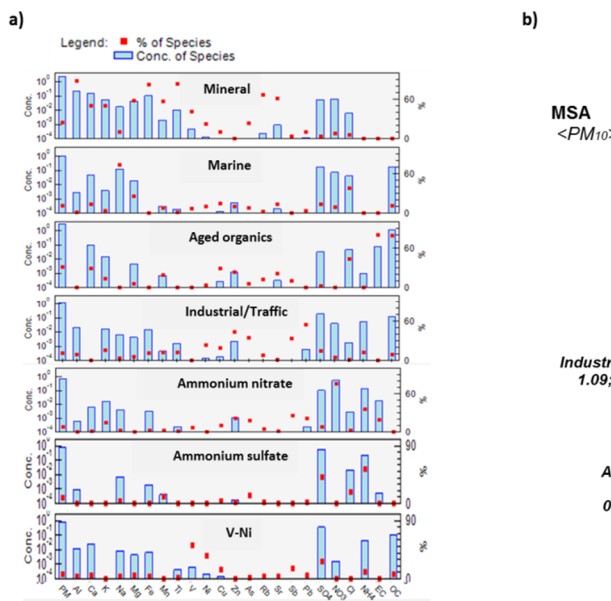

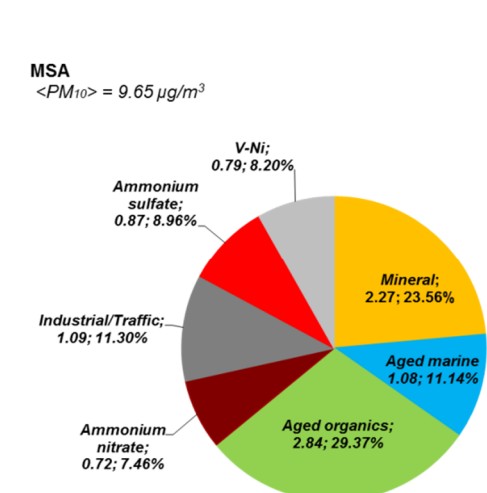

**Figure 2**





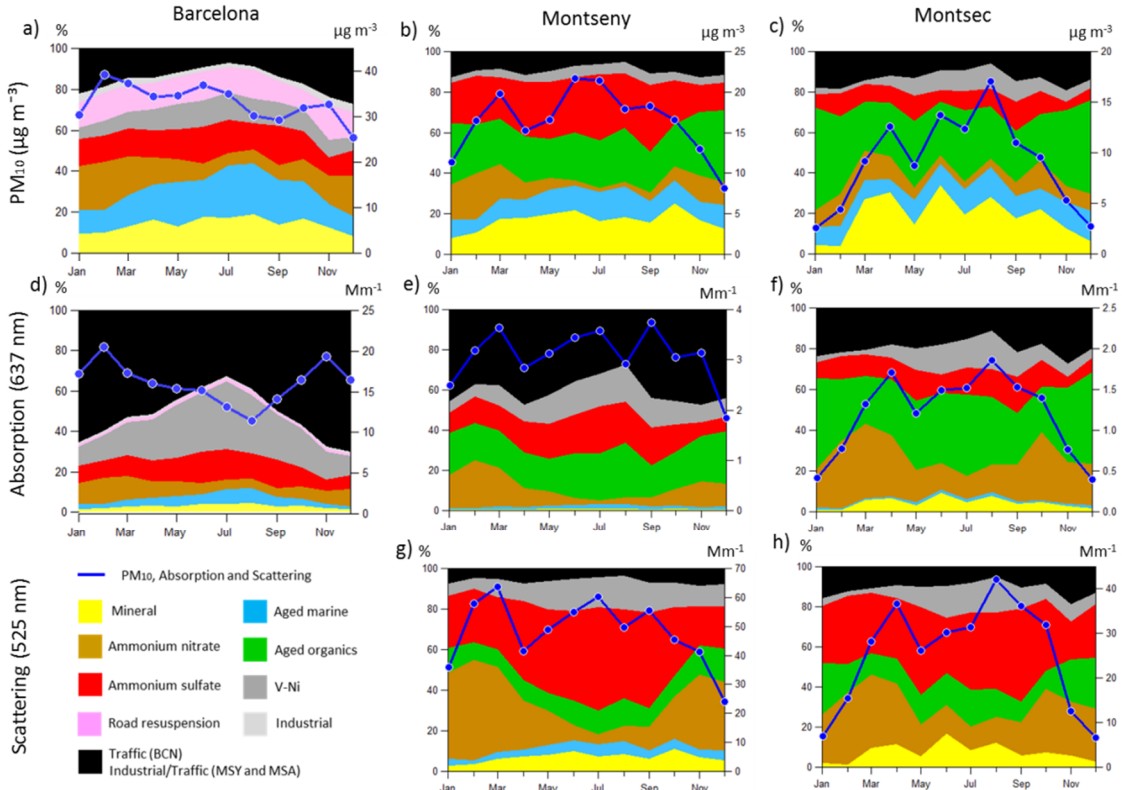

**Figure 3**




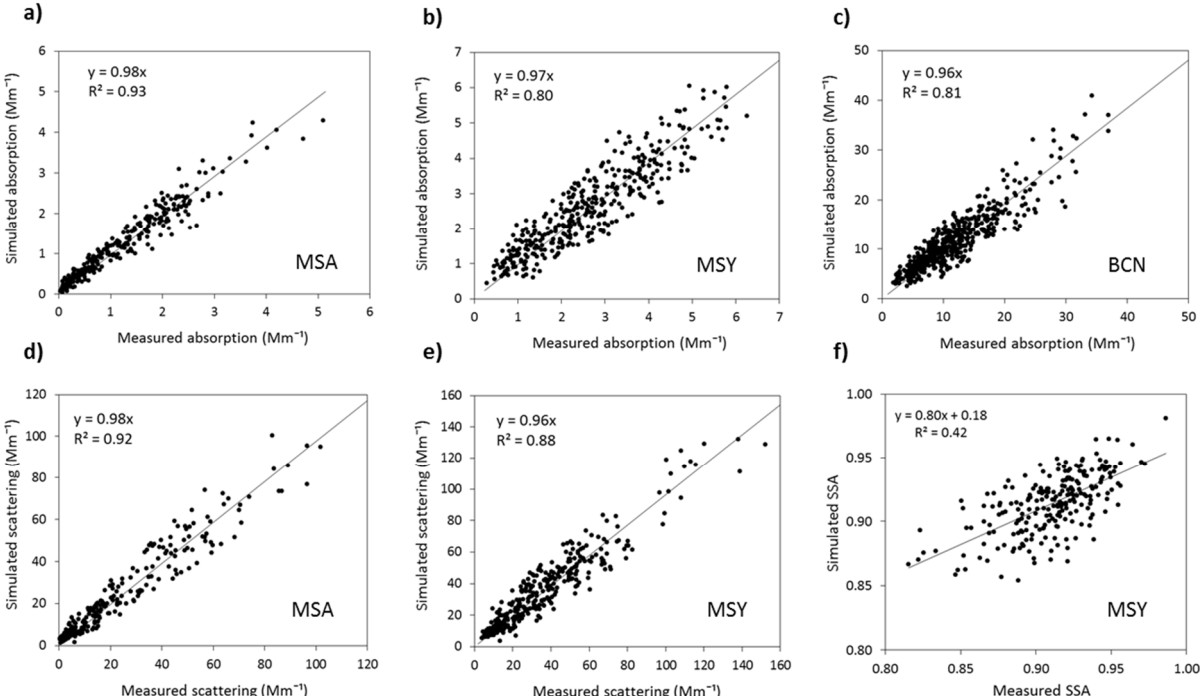

**Figure 4**



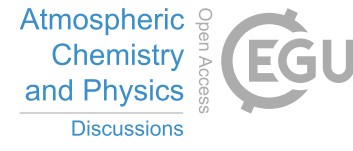

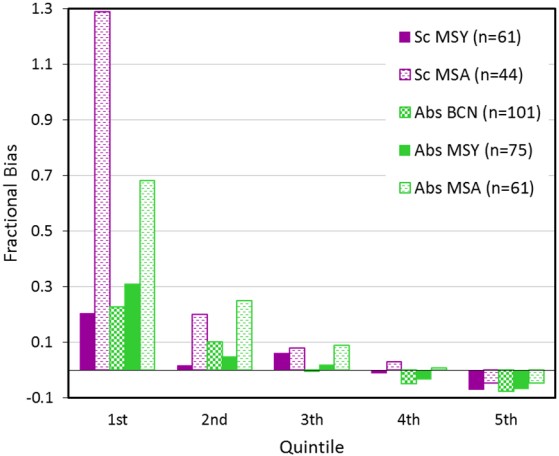

**Figure 5**



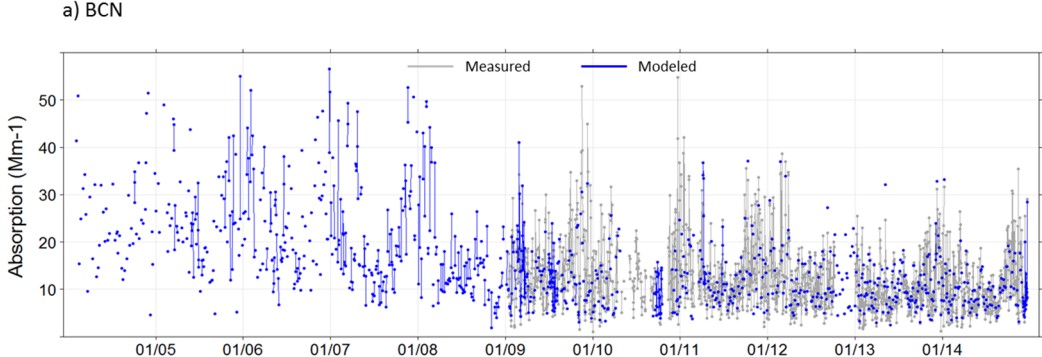

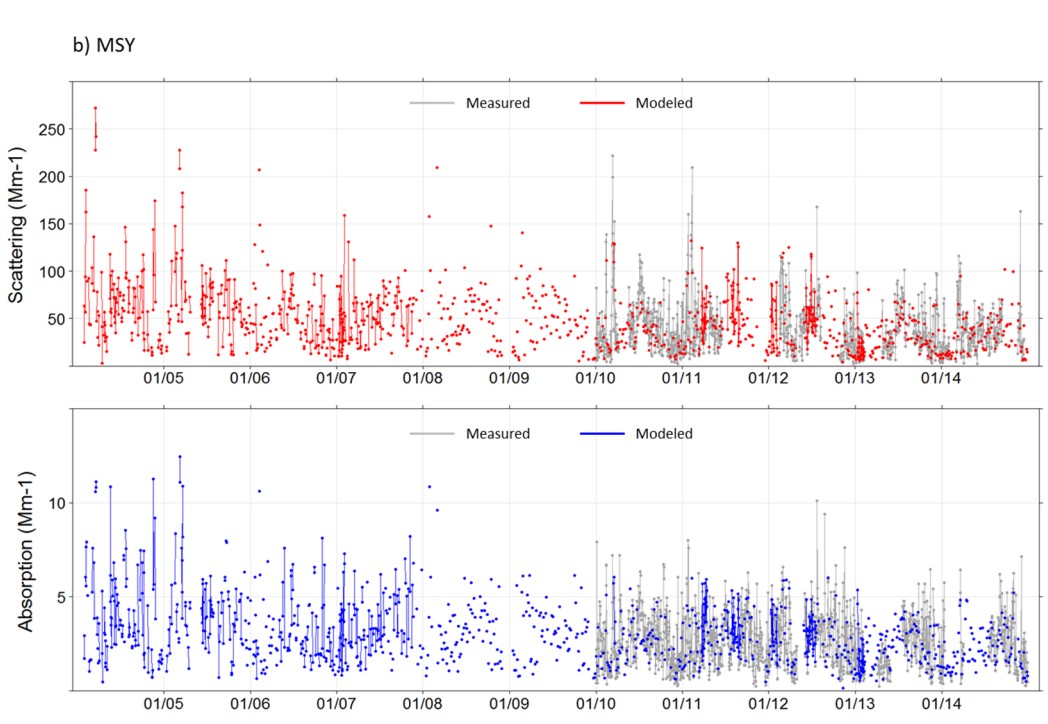





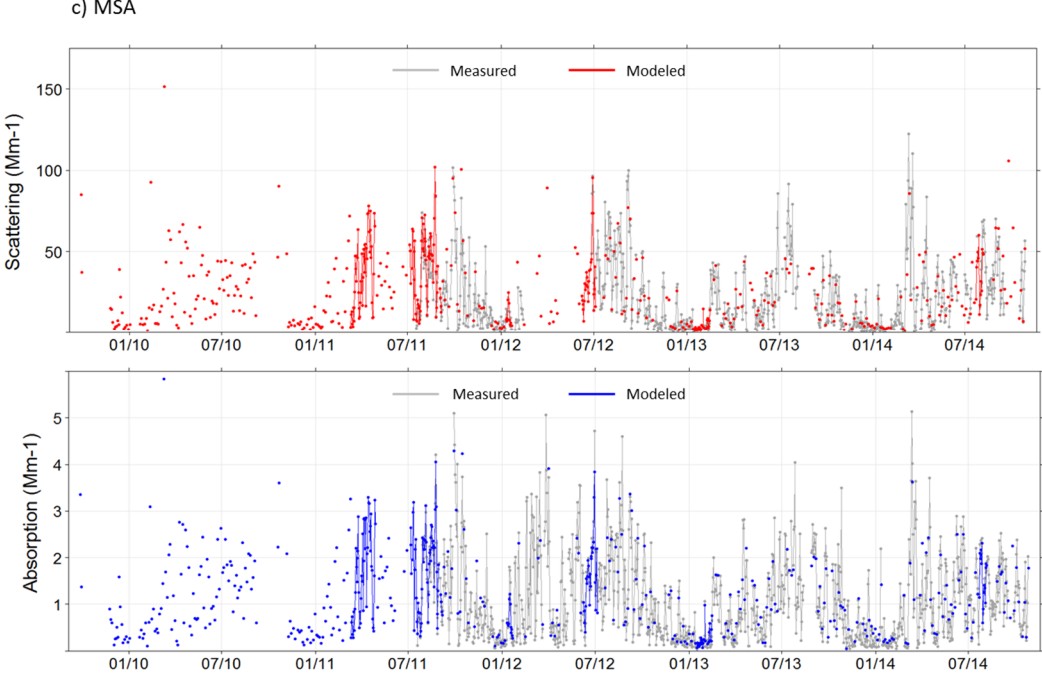

**Figure 6**





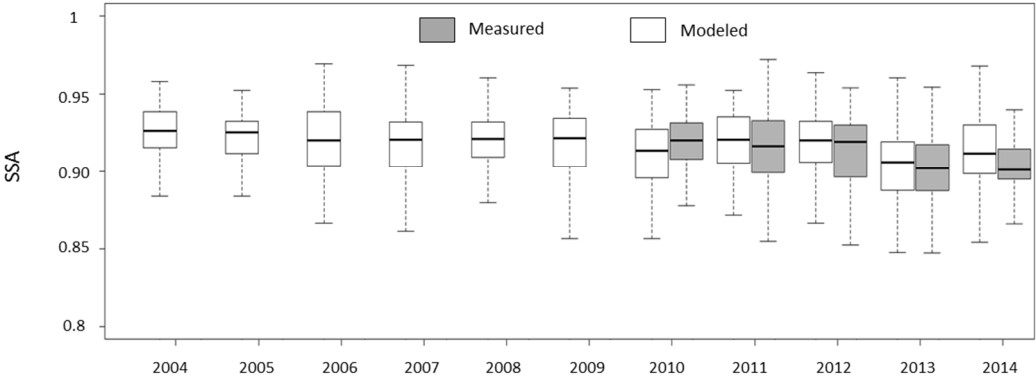

**Figure 7**





Figure 8