# Peer review of "Impact of aerosol particle sources on optical properties at urban, regional and remote levels in the north-western Mediterranean"

_Atmospheric Chemistry and Physics, 2017_

## Referee Comment (RC1) · Anonymous Referee #1 · 3 May 2017

Overall Quality/General Comments:

The primary objective of the paper is to quantify the mass scattering and absorption efficiencies (MSE and MAE) of different aerosol source types observed at urban (Barcelona-BCN; MAE only), regional background (Montseny-MSY), and remote background (Montsec-MSA) sites in the northwestern (NW) Mediterranean region. To accomplish this, the authors applied a multi-linear regression model (MLR; Eqs. 1 and 2) to a ∼4-year time series of aerosol source type mass concentrations (derived using PMF), PM10 aerosol light scattering and absorption coefficients ($\sigma$sp and $\sigma$ap). The

aerosol source mass concentrations served as dependent variables and the $\sigma$sp and $\sigma$ap served as independent variables. Note that I am using the common symbols $\sigma$sp and $\sigma$ap as shorthand for the coefficients. The authors did not use these symbols. The combined PMF/MLR approach yields more useful MSE and MAE for source apportionment studies, since they are given in terms of aerosol source types instead of chemical components. Armed with the SAE and MAE for each of the observed aerosol source types, the authors calculated the $\sigma$sp and $\sigma$ap contributions for each source type and summed the contributions to yield total calculated $\sigma$sp and $\sigma$ap. They then compared these calculated coefficients with measured $\sigma$sp and $\sigma$ap (at RH$\leq$40%) over the period (2010-2014 at BCN and MSY; 2011-2014 MSA) during which aerosol optical properties and composition were simultaneously measured. They used good agreement in this comparison to validate their source-specific SAE and MAE. They then used the source-specific SAE and MAE along with measured composition to reconstruct $\sigma$sp and $\sigma$ap for pre-2010, when there were only mass concentration measurements (i.e. no $\sigma$sp and $\sigma$ap measurements). The reconstructed from $\sigma$sp and $\sigma$ap was merged with the measured values from 2010-2014 at MSY in a trend analysis. The combined PFM//MLR technique could find use in other source apportionment studies and the derived MAE and MSE for the individual aerosol sources also find potential use in the modeling community (although I am not a modeler) and possibly in regional pollution mitigation strategies. However, I believe that serious issues regarding scientific quality and presentation quality must be fixed before this manuscript is acceptable for publication. These issues are described in the broad and specific comments below.

Scientific Significance: The perceived significance lies in the fact that the authors derived MSE and MAE for the various aerosol source types, instead of doing so for speciated mass concentrations. The PFM/MLR technique yielding source-specific MSE and MAE is novel (to the best of my knowledge) and the results of Sect(s). 3.1-3.4 could contribute to improved knowledge of source-apportioned contributions to aerosol light scattering and absorption coefficients over the 2010-2014 period at the 3 NW Mediterranean sites (minus the MAE at BCN). The utility of the reconstructed $\sigma$sp and $\sigma$ap,

and single-scattering albedo (SSA) for pre-2010 (Sect. 3.5) and the resulting trend studies (Sect.3.6) is questionable-in my opinion. See my comments below regarding Scientific Quality. I would rate the Scientific Significance as 'good' if the authors focus on the results of Sect(s). 3.1-3.4 and improve upon the Scientific Quality.

Scientific Quality: I rate the scientific quality as 'fair'. I am not an expert on PMF but the scientific approach and applied methods seem acceptable, up until Sect. 3.5-Reconstruction of scattering, absorption, and SSA time series. In Sect. 3.5, the authors discuss strong correlations between the measured and calculated aerosol $\sigma$sp , $\sigma$ap, and SSA for the period 2010-2014 at BCN and MSY, and for the period 2011-2014 at MSA, when the optical property measurements are available (Fig. 4). They state (bottom of P.16) that "As a result, long-term time series of scattering and absorption were satisfactory reconstructed when chemical data was available, for the period 2004-2014 at BCN and MSY and for the period 2011-2014 at MSA (Fig. 6)." However, high correlation and good agreement between measured and calculated scattering and absorption coefficients is to be expected, since the since the source-specific MSE and MAE are being evaluated using the same dataset that was used to determine them (via the MLR). The authors could (should?) have used different subsets of the period to test the model than that used to develop the model. The utility of the reconstructed SSA is highly questionable. In addition to the issue that I just discussed, the agreement between measured and calculated SSA was marginal during the 2010-2014 period at MSY (R2=0.42; slope=0.80). This marginal agreement does not inspire confidence that the reconstructed SSA is sufficiently accurate for use in any trend studies (Sect. 3.6), much less the assertions made by the authors regarding the results from this trend study. There are also many cases where the claims made by the authors are not supported by the available data (See my specific comments below) or references to the sources of claims made by the authors are not given.

Presentation Quality: This is the major weakness of the paper and requires significant improvement before the paper is acceptable for publication. I would rate this as 'poor'

for the current version of the manuscript. My major criticisms are related to grammar and (to a lesser degree) structure of some paper sections (mainly the Introduction). The manuscript is full of long, rambling sentences with incorrect punctuation (ex: missing commas), misspellings, and improper usage of tenses. The first paragraph of the Introduction section is a good example of the above-mentioned grammatical errors but they pervade throughout the paper. The grammar also limited my ability to understand some of the authors' interpretation of results. In several places, the wording likely does not convey their meaning (see specific comments for a few examples). There are way too many grammatical errors for me to list in my specific comments and doing so detracts my focus from the other aspects of the paper. I strongly encourage the authors to have someone (colleague or an English editor) carefully look over the document for grammar and fix this. Then have the same colleague or someone else carefully evaluate the grammar of the revised manuscript. Related to grammar is the inconsistent use of tenses. Past and present tenses are used interchangeably when describing methods and results reported by others. Past tense should be consistently used to describe previous studies and present tense is typically used to describe the current study. Please fix this throughout the manuscript. The structure of Introduction section should also be modified to improve readability and a few figures require improvements (See my specific comments below regarding these issues).

In summary, I would rate the current version of the manuscript as 'fair'.

Specific Comments and Technical Corrections:

Note that there are too many grammatical issues for me to list so I only listed a few. Please have a colleague review the manuscript very carefully for grammar and then fix.

- The abstract is too long and should be shortened.

- Readability of the Introduction section would be much improved if the authors arranged it in the following order: (1) Statement of the problem-Why is knowledge of MAE and MSE important? (2) Results from previous works; (3) How this study will

advance knowledge and what is unique about it? These are all included in the existing version of the Introduction but are mixed, long-winded, and the section does not read well. Significantly reduce description of methods (both those of others and the current study) to only that necessary to accomplish (1)-(3) and save any more details for the Methodology section.

- P.3 Lines 26-32: The wording of temporal and spatial aerosol variability and the reasons for this variability is repetitive and this 7-line passage could be condensed into ∼3 lines

- P.4 Line 1: "The determination of MSE and MAE for specific aerosol components has been subject of research in the last few years." It has been the subject of research for more than the last few years so this sentence should be reworded or omitted. My suggestion is to omit it, as it is vague and really does not add much to the paragraph.

- P.6 Lines 19-24: "The three sites are involved in the Catalonian air quality monitoring network. Additionally, the MSY and MSA stations form part of the ACTRIS (Aerosol, Clouds and Trace gases Research InfraStructure) and GAW (Global Atmosphere Watch) networks, and then aerosol optical measurements were performed following the standards required by these networks." I suggest re-wording as "The three sites are members of the Catalonian air quality monitoring network, ACTRIS (Aerosol, Clouds and Trace gases Research InfraStructure) and GAW (Global Atmosphere Watch). Aerosol optical properties at the sites are measured following standard network protocols". Then include reference(s) describing these protocols.

- P.7 Line 7: "Samples were collected every 3/4 days". I am confused as to whether this means "3/4 day" or "3 to 4 days". I am assuming that the authors imply the former but then remove the 's' from 'days'. This also illustrates the author' usage of past tense to describe the current study. I recommend consistent usage of present tense for this throughout the manuscript.

- P.7 Line 7: Include a reference for the protocol. There are MANY other instances in

the paper where the authors make mention of protocols without providing references.

- P.9 Lines 3-6. "It is remarkable that differences in the sampling conditions (RH, size cut) or chemical analysis methods influence the resulting efficiencies obtained for different emplacements. In this study, scattering RH was controlled below 40% preventing the hygroscopic growth of the particles, which could lead to an enhancement in the scattering efficiency." Please clarify why this is remarkable or else delete or rephrase the sentence. The following sentence is also not true and should be modified or removed. Just because RH<40% does not prohibit scattering enhancements, especially for organics. The water uptake is small (on the order of ∼10% growth) but not prohibited. Size cut also influences aerosol intensive properties, including single-scattering albedo, Angstrom exponent, etc. This can be seen from any of the papers based on measurements at the NOAA-GMD monitoring sites (Sheridan et al., 2001; Delene and Ogren, 2002; Sherman et al., 2015; Andrews et al., 2011; . . . ...).

- P.10 Line 7: Change 'PM10 levels" to '"PM10 mass concentrations".

- P.10 Line 10: Change "PM10 load." To '"PM10 mass concentrations". Please do this for all instances in the manuscript.

- Fig.3b. PM10 mass concentration at Montseny is nearly as high in March as in June-July and is higher than in August but this is not discussed at all.

- Fig(s).4. Please use site abbreviations as either as sub-plot titles or legend labels in fig(s) 4 to make it easier to look at the figures, without needing to go back and forth between plots and caption to see which plot corresponds to which site.

- P.12 Lines 4-5: The assertion that "Aged organics at MSA (29%) and Ammonium sulfate at MSY (24%) were the dominant sources throughout the year and reached the largest absolute contribution in summer" is not supported by Fig.3. The Mineral source at MSA is comparable to Aged Organics during several months of spring/summer and organics are equal to or exceed ammonium sulfate for several months at MSY. Please

reword this assertion to better reflect the data in Fig.3.

- P.12 Lines 14-16. This has already been mentioned more than once so the sentence should be deleted.

- P.12 Line 12. Add the word 'Mass' to the beginning of Sect. 3.3 title and to 'absorption efficiencies' and 'scattering efficiencies' throughout this section.

- P.12 Lines 17-24. This passage has already been discussed in previous sections and is not a result. Therefore, it should be deleted.

- P.13 Line 2: Change the word 'coefficient' to 'MSE'.

- P.13 Lines 31-32: "Interestingly, a higher scattering wavelength dependence was observed for those sources with higher contribution from anthropogenic tracers which are mainly present in the fine mode (Table 2)." This is to be expected. Size distributions with higher contributions from the fine mode will possess larger variation of scattering coefficient with wavelength than size distributions with larger contributions from coarse-mode aerosol.

- P.14 Line 19: Please clarify what you mean by "European scenarios".

- P.15 Lines 9-12:" Both sources presented inverse seasonal cycles following the seasonal variation of mass contributions, with Ammonium sulfate maximizing in summer at MSY (46%) whereas showing similar contribution throughout the year at MSA. Conversely, Ammonium nitrate mainly governed the light scattering in winter (42% and 29% at MSY and MSA)." Please change the wording of the first sentence, as the phrase 'inverse seasonal cycles' is not clear. Wording similar to this is used in other places to describe the cycles and should be fixed. It is clearer to simply state something along the lines of "The annual cycles of ammonium sulfate and ammonium scattering coefficients follow those of the PM10 mass concentration, with summer maxima and winter minima". The assertion that there are similar ammonium sulfate contributions throughout the year at MSA is not supported by Fig.3h, which indicates that the fraction of

light scattering attributed to ammonium sulfate is highest in Aug-Sept and lowest in Nov-Dec.

- P.15 Lines 15-17: "Light absorption appeared to be almost dominated by the Traffic source at BCN and in a minor proportion by the equivalent Industrial/Traffic at MSY and MSA (Fig. 3d, e, f), showing high contributions in winter (65%, 42%, 22%) despite the relative low mass concentration (23%, 11%, 17%)." This is one of many instances throughout the paper where the wording probably does not convey the authors' intentions. The meaning of 'almost dominated' is unclear. Light absorption coefficient is either dominated by Industrial/Traffic or it is not. Based on Figs.3d-3e, it looks as if it is only dominated by Industrial/Traffic during non-summer months at BCN, although it clearly exerts the major influence (∼40%) at MSY. I also do not understand the meaning of "a minor proportion by the equivalent Industrial/Traffic at MSY and MSA". The influence of Industrial/Traffic (∼40-50% at MSY and ∼20% at MSA) is neither dominating or minor. Please clarify this and similar statements throughout the document so that their meaning reflects your intentions. You do a much better job on P.15 Lines 30-31.

- P.15 Lines 20-21: "Therefore, Traffic, Industrial/Traffic and V-Ni sources which highly influence air quality also have caused an important effect on radiative forcing, particularly in those sites closer to the emission sources." This assertion may be true but cannot be supported, given the measurements available in the current study. Only absorption coefficient data is available at BCN (the main site influenced by these sources) and radiative forcing depends primarily on aerosol optical depth (which in turn is largely controlled by scattering coefficient). As such, I recommend that the authors either remove this statement or provide more support for it, given the available data.

- P.15 Line 26-P.16 Line 3: A summary of results should be placed in the Summary and Conclusions section, not in the Results section.

- P.15 Lines 26-27: "As a summary, we have shown that the main target pollutant

sources affecting air quality degradation have caused important effects on light extinction in the northwestern Mediterranean." This sentence should be clarified. You cannot state anything regarding light extinction at BCN without measurements of light scattering coefficient, which represents the major contribution to light extinction coefficient.

- P.16 Line 1: Please remove the phrase "As a novelty,".

- P.16 Line 19: What 'published results'? Please cite reference(s) to support this.

- P.17 Line 4: State at which site you are discussing the SSA. You state that it is MSY in the caption of Fig.4 but it should also be stated in your discussion, so that the reader does not need to go back and forth between the discussion and figure captions.

- P.17 Line 5: The slope of modeled versus measured SSA is 0.80, which is not close to 1. Please reword this.

- P.17 Line 15: Since trend study was only done for MSY, the section title should be rephrased to more accurately represent the section contents. This could be as simple as "Long-term trends in scattering and absorption coefficients at MSY".

- P.17 Lines 19-21: "Despite a larger uncertainty was found for the modeled SSA, this technique allowed to further investigate the temporal trend of this important parameter and its relation with changes in atmospheric composition (Fig. 7)." Marginal agreement of measured and model SSA ($R^2$=0.42, slope =0.80) during the 2010-2014 period at MSY (when agreement should be best, since the authors use the same period to both generate and evaluate the model) likely makes it impossible to state much about any long-term trends in SSA at MSY. As such, I believe that the discussion of SSA trends at MSY should be removed from the paper.

- P. 18 Lines 4:6:" A marked decline was also observed for nitrate and sulfate PM in other European monitoring sites since 1990, as outlined 5 in the EMEP report 1/2016." Please provide a reference for the EMEP report. Please do the same for other instances where sources are named but not referenced.

- P.18 Lines 7-17: Please cite references for all of the claims made in this paragraph, since they are not supported by the data presented in this manuscript. Were the claims from the Pandolfi et al. (2016) paper? If so, you should state this and probably not spend a paragraph summarizing a different study. Simply state (as you did in first sentence of the paragraph) that the causes of reductions in most sources at MSY are discussed in Pandolfi and then move on to how your study contributes to the knowledge. If not, please provide the sources to justify the assertions.

- P.18 Lines 22-23: "Interestingly, the SSA showed a significant decreasing trend of -0.11 % y-1 (-0.001 yr-1) leading to a total reduction (TR) of -1.24 % since 2004 at MSY, pointing that the atmosphere is getting significantly darker." A change in SSA of 0.01 is relatively small and does not imply that the atmosphere is getting significantly darker.

- P. 18 Lines 25-28: "Differences in the SSA reduction at both sites might be explained by the severe pollution episodes taking place in the Po Valley resulting in a higher dominance of absorption in the light extinction process, compared to MSY which is representative of a less polluted environment." You should provide some reference for this. If not, the decreasing SSA could just as easily be due to larger reductions in scattering than absorption.

- P.18 Lines 29-34: This is another example of claims being made (and percentages brought up) without any references for these numbers or results. Please include sources/references for these numbers.

- P.19 Lines 12-13: "The reduction in the SSA trend points out the increasing prominence of absorption in the light extinction process in the NW Mediterranean." This represents another claim that is unsupported by the data. An apparent 1% decrease in SSA at a single site does NOT point to increasing prominence of absorption in the light extinction process in the NW Mediterranean region. Please either provide more support for this assertion or delete it.

- P. 19 Lines 16-23: This discussion of radiative forcing and relation to policy lies well beyond what can be stated, based on data/results from this manuscript. Please either change the wording so that it is supported by your results or else remove.

- P. 20 Lines 31-32: "A total reduction (TR) of -1.12% in the SSA was mainly motivated by the heterogeneous and non-selective reduction of key aerosol sources showing opposite effects on radiative forcing." The assertion regarding effects on radiative forcing is not supported by the data and should be removed.

- P.20 Line 34-Page 21 Line 8: "However, these measures have resulted in a more pronounced reduction of light-scattering aerosol sources (Ammonium nitrate, Ammonium sulfate), leading to an increase of the incoming solar radiation and therefore contributing to climate warming. This positive radiative effect is enhanced by the less effectiveness of air quality strategies for reducing light-absorbing sources containing dark particles. A decrease in the SSA trend points to a darkening of the atmosphere and consequently to a progressive predominance of absorption in the light extinction process in the NW Mediterranean. Accordingly to the results presented in this work, future strategies need to focus on preferentially reducing atmospheric aerosols mainly originated 5 from combustion sources. Industrial/Traffic and V-Ni aerosol sources, which highly contributed to air quality degradation but also to light absorption, should be abated thus addressing win-win policies aimed to improve air quality and mitigate climate warming in the NW Mediterranean."

Where is the evidence for increasing in solar radiative and contributions to climate warming? This and the policy recommendations are completely unsubstantiated by the data and either need to be justified or removed. A small and questionable decrease in SSA (based on poor correlation and marginal agreement between modeled and measured SSA during common period) at a single site cannot be used to make claims regarding darkening of the atmosphere. See my many comments above regarding assertions that are unsupported.

---

## Referee Comment (RC2) · Anonymous Referee #2 · 6 Jun 2017

This study showed not only aerosol source contributions based on a positive matrix factorization (PMF) analysis, but also mass scattering and absorption efficiencies (MSE and MAE) of different aerosol sources by using multilinear regression method at urban, regional and remote backgrounds in the Spain. Although the results and discussion were documented well, there are several important shortcomings.

• While the detailed pedagogical description of the approach is appreciated, the paper overall must be shortened. There are many repetitions in the text or not essentially needed with many references. This review strongly suggest that text must be made

more succinct. For example, in INTRODUCTION, the authors explained overall aspects of aerosols in climate-air quality research. Most of them are overstated and not directly related with the results of this study. The words "From air quality to climate" in title also should be removed. Summary and conclusions also should be shortened; highlight the major findings succinctly and provide directions/implications of this work.

• Generally, source apportionment by PMF model shows the contributions of aerosol chemical properties from various sources, such as traffic (vehicle), biomass burning, dust (road dust), marine, industry, secondary nitrate, secondary sulfate, ship, etc. However, the sources given in this study are aerosol compositions, except for industrial/Traffic and Marine. Firstly, the authors should show the chemical compositions and discuss the characteristics during the study period. The explanation of major emissions sources of the aerosols also provided. Secondly, more detailed descriptions for source profiles should be given. For example, V-Ni at MSY originated mainly form shipping emissions (see section 3.3), why EC is not considered at the source profile.

• It is not fully explained how the authors determined the scattering and extinction efficiencies (MSE and MEE) of chemical species by using the Multilinear regression method. Eqs. 1 and 2 are not enough. This reviewer strongly suggests that detailed steps for getting the MSE and MEE of chemical species by using the multilinear regression have to be present in main text or in the supplement.

• Regarding to the reconstruction (section 3.5), what is the root-mean-square difference for data given in Figure 4? This reviewer suggests to use percentage difference of bias rather than fractional bias.

• Long-term trends in Section 3.6: There is no detailed explanation on Figure 6. There is a critical issue that how we can trust these reconstructed values given in Figure 6 and associated trend analysis results. For example, Collaud Coen et al (http://www.atmos-chem-phys.net/13/869/2013/acp-13-869-2013.pdf) reported that, in the Europe, significant trends were not observed for aerosol optical properties at most

sites (see Table 8 in the paper). However, this study, this study reports about a scattering TR(total reduction) of -52% since 2004 and -44% in the absorption coefficient. Therefore, the authors should investigated all published literature, especially in Europe, and give a reasonable and reliable explanation with evidences. If there are such distinct decreases in aerosol scattering and absorption properties, is this trend also consistently seen from aerosol optical depth measurements from AERONET/SKYNET and satellite sensors? Also, do PM2.5 concentrations show a decreasing trend over the study period?

---

## Author Comment (AC1) · 9 Oct 2017

**Impact of aerosol sources on optical properties at urban, regional and continental levels in the north-western Mediterranean**

The authors would like to thank the reviewers for their comments and suggestions, which helped improving the quality of this work. A new version of the manuscript has been prepared following the referee's suggestions. We provide below detailed replies to each of the comments.

Anonymous referee #1

Overall Quality/General Comments: The primary objective of the paper is to quantify the mass scattering and absorption efficiencies (MSE and MAE) of different aerosol source types observed at urban (Barcelona-BCN; MAE only), regional background (Montseny-MSY), and remote background (Montsec-MSA) sites in the northwestern (NW) Mediterranean region. To accomplish this, the authors applied a multi-linear regression model (MLR; Eqs. 1 and 2) to a ~4-year time series of aerosol source type mass concentrations (derived using PMF), PM10 aerosol light scattering and absorption coefficients ($\sigma sp$ and $\sigma ap$). The aerosol source mass concentrations served as dependent variables and the $\sigma sp$ and $\sigma ap$ served as independent variables. Note that I am using the common symbols $\sigma sp$ and $\sigma ap$ as shorthand for the coefficients. The authors did not use these symbols. The combined PMF/MLR approach yields more useful MSE and MAE for source apportionment studies, since they are given in terms of aerosol source types instead of chemical components. Armed with the SAE and MAE for each of the observed aerosol source types, the authors calculated the $\sigma sp$ and $\sigma ap$ contributions for each source type and summed the contributions to yield total calculated $\sigma sp$ and $\sigma ap$. They then compared these calculated coefficients with measured $\sigma sp$ and $\sigma ap$ (at RH≤40%) over the period (2010-2014 at BCN and MSY; 2011-2014 MSA) during which aerosol optical properties and composition were simultaneously measured. They used good agreement in this comparison to validate their source specific SAE and MAE. They then used the source-specific SAE and MAE along with measured composition to reconstruct $\sigma sp$ and $\sigma ap$ for pre-2010, when there were only mass concentration measurements (i.e. no $\sigma sp$ and $\sigma ap$ measurements). The reconstructed from $\sigma sp$ and $\sigma ap$ was merged with the measured values from 2010-2014 at MSY in a trend analysis. The combined PFM//MLR technique could find use in other source apportionment studies and the derived MAE and MSE for the individual aerosol sources also find potential use in the modeling community (although I am not a modeler) and possibly in regional pollution mitigation strategies. However, I believe that serious issues regarding scientific quality and presentation quality must be fixed before this manuscript is acceptable for publication. These issues are described in the broad and specific comments below. Scientific Significance: The perceived significance lies in the fact that the authors derived MSE and MAE for the various aerosol source types, instead of doing so for speciated mass concentrations. The PFM/MLR technique yielding source-specific MSE and MAE is novel (to the best of my knowledge) and the results of Sect(s). 3.1-3.4 could contribute to improved knowledge of source-apportioned contributions to aerosol light scattering and absorption coefficients over the 2010-2014 period at the 3 NW Mediterranean sites (minus the MAE at BCN). The utility of the reconstructed $\sigma sp$ and $\sigma ap$, and single-scattering albedo (SSA) for pre-2010 (Sect. 3.5) and the resulting trend studies (Sect.3.6) is questionable-in my opinion. See my comments below regarding Scientific Quality. I would rate the Scientific

**Significance as 'good' if the authors focus on the results of Sect(s). 3.1-3.4 and improve upon the Scientific Quality.**

**Reviewer#1. General comment 1. Scientific Quality: I rate the scientific quality as 'fair'. I am not an expert on PMF but the scientific approach and applied methods seem acceptable, up until Sect. 3.5- Reconstruction of scattering, absorption, and SSA time series. In Sect. 3.5, the authors discuss strong correlations between the measured and calculated aerosol σsp , σap, and SSA for the period 2010-2014 at BCN and MSY, and for the period 2011-2014 at MSA, when the optical property measurements are available (Fig. 4). They state (bottom of P.16) that "As a result, long-term time series of scattering and absorption were satisfactory reconstructed when chemical data was available, for the period 2004- 2014 at BCN and MSY and for the period 2011-2014 at MSA (Fig. 6)." However, high correlation and good agreement between measured and calculated scattering and absorption coefficients is to be expected, since the since the source-specific MSE and MAE are being evaluated using the same dataset that was used to determine them (via the MLR). The authors could (should?) have used different subsets of the period to test the model than that used to develop the model.**

Authors agree with the referee, independent datasets of source mass contributions together with the obtained source specific MSE/MAE, should have been used to reconstruct scattering (σsp) and absorption (σap) coefficients for a different period not used in the MLR analysis.

This analysis was not performed in the initial version of the manuscript due to the limited availability of chemical analysis results for ions concentration (Cl⁻, NO₃⁻, SO₄) for the year 2015.

In the revised paper we present reconstructed time series of σsp and σap for the year 2015 at MSY. With this aim, a new PMF was performed for the period 2004-2015. Then, the resulting PMF source contributions obtained for the period January-December 2015 were used together with the corresponding MSE and MAE, previously obtained by means of the MLR model for the period 2010-2014, in order to compute σsp and σap coefficients for the year 2015.

This analysis provides an insight on the capability of the PMF-MLR technique to estimate σsp and σap coefficients for those periods when optical properties measurements are not available.

Following the Referee's suggestion, section 3.5 of the revised manuscript contains the evaluation of the PMF-MLR technique using independent datasets of chemical source contributions for the year 2015. The following paragraph has been added to the revised paper:

"An independent subset of the study period was considered, in order to further evaluate the PMF-MLR technique and the accuracy of the method to simulate optical properties when chemical source contributions were available. Therefore, a new PMF was performed in order to obtain the source contributions for the period 2004-2015 at MSY. With this aim, the simulation of $\sigma_{sp}$ and $\sigma_{ap}$ coefficients for the period January-December 2015 was carried out by means of the source specific MSE and MAE previously obtained in the MLR analysis for the period 2010-2014. Good agreement was found between modeled and measured $\sigma_{sp}$ ($R^2$=0.85) and $\sigma_{ap}$ ($R^2$=0.76) coefficients, at 525 and 637 nm respectively, showing slopes close to one for the year 2015 at MSY (Fig. 6). This analysis confirms the confidence of the PMF-MLR technique to accurately estimate $\sigma_{sp}$ and $\sigma_{ap}$ coefficients when chemical data is available."

[Figure]

Figure 5. Relationship between measured and modeled (a) scattering at 525 nm and (b) absorption at 637 nm at MSY for the period January 2015-December 2015.

**Reviewer#1. General comment 2). The utility of the reconstructed SSA is highly questionable. In addition to the issue that I just discussed, the agreement between measured and calculated SSA was marginal during the 2010-2014 period at MSY (R2=0.42; slope=0.80). This marginal agreement does not inspire confidence that the reconstructed SSA is sufficiently accurate for use in any trend studies (Sect. 3.6), much less the assertions made by the authors regarding the results from this trend study. There are also many cases where the claims made by the authors are not supported by the available data (See my specific comments below) or references to the sources of claims made by the authors are not given.**

Following the referee's comment, the SSA trend analysis has been removed from section 3.6. Results making reference to this part of the paper have also been removed from the conclusions section. Modifications are directly shown in the revised manuscript.

**Reviewer#1. General comment 3). Presentation Quality: This is the major weakness of the paper and requires significant improvement before the paper is acceptable for publication. I would rate this as 'poor' for the current version of the manuscript. My major criticisms are related to grammar and (to a lesser degree) structure of some paper sections (mainly the Introduction). The manuscript is full of long, rambling sentences with incorrect punctuation (ex: missing commas), misspellings, and improper usage of tenses. The first paragraph of the Introduction section is a good example of the above-mentioned grammatical errors but they pervade throughout the paper. The grammar also limited my ability to understand some of the authors' interpretation of results. In several places, the wording likely does not convey their meaning (see specific comments**

**for a few examples). There are way too many grammatical errors for me to list in my specific comments and doing so detracts my focus from the other aspects of the paper. I strongly encourage the authors to have someone (colleague or an English editor) carefully look over the document for grammar and fix this. Then have the same colleague or someone else carefully evaluate the grammar of the revised manuscript. Related to grammar is the inconsistent use of tenses. Past and present tenses are used interchangeably when describing methods and results reported by others. Past tense should be consistently used to describe previous studies and present tense is typically used to describe the current study. Please fix this throughout the manuscript. The structure of Introduction section should also be modified to improve readability and a few figures require improvements (See my specific comments below regarding these issues). In summary, I would rate the current version of the manuscript as 'fair'.**

Following the Referee's suggestions regarding presentation quality of the manuscript, grammatical aspects of the discussion paper have been revised for a better understanding. Particular attention has been paid to the proper use of tenses and the structure of the introduction section. Changes have been directly modified in the discussion paper.

**Specific Comments and Technical Corrections: Note that there are too many grammatical issues for me to list so I only listed a few. Please have a colleague review the manuscript very carefully for grammar and then fix.**

**Reviewer#1. Specific comment 1). The abstract is too long and should be shortened.**

The abstract has been shortened from 574 to 453 words. Modifications are directly included in the revised manuscript.

**Reviewer#1. Specific comment 2). Readability of the Introduction section would be much improved if the authors arranged it in the following order: (1) Statement of the problem-Why is knowledge of MAE and MSE important? (2) Results from previous works; (3) How this study will advance knowledge and what is unique about it? These are all included in the existing version of the Introduction but are mixed, long-winded, and the section does not read well. Significantly reduce description of methods (both those of others and the current study) to only that necessary to accomplish (1)-(3) and save any more details for the Methodology section.**

The introduction section has been rephrased. Modifications have been directly added to the revised manuscript.

**Reviewer#1. Specific comment 3). P.3 Lines 26-32: The wording of temporal and spatial aerosol variability and the reasons for this variability is repetitive and this 7-line passage could be condensed into ~3 lines.**

The paragraph has been shortened from 128 to 71 words:
"However, a thorough quantification of the direct and indirect aerosol effects on the Earth's radiative budget is difficult to achieve (Zieger et al., 2012). The high spatial and temporal variability of atmospheric aerosols along with the large differences in particle composition and size (Andrews et al., 2011; Bond et al., 2013; Haywood et al., 1999), results in a changing radiative forcing from local to global scales (Collaud Coen et al., 2013)."

**Reviewer#1. Specific comment 4. P.4 Line 1: "The determination of MSE and MAE for specific aerosol components has been subject of research in the last few years." It has been the subject of research for more than the last few years so this sentence should be reworded or omitted. My suggestion is to omit it, as it is vague and really does not add much to the paragraph.**

Following the referee's suggestion, the sentence has been removed from the revised manuscript.

**Reviewer#1. Specific comment 5****. P.6 Lines 19-24: "The three sites are involved in the Catalonian air quality monitoring network. Additionally, the MSY and MSA stations form part of the ACTRIS (Aerosol, Clouds and Trace gases Research InfraStructure) and GAW (Global Atmosphere Watch) networks, and then aerosol optical measurements were performed following the standards required by these networks." I suggest re-wording as "The three sites are members of the Catalonian air quality monitoring network, ACTRIS (Aerosol, Clouds and Trace gases Research InfraStructure) and GAW (Global Atmosphere Watch). Aerosol optical properties at the sites are measured following standard network protocols". Then include reference(s) describing these protocols.**

It should be noted that the Barcelona site is not a member of the ACTRIS/GAW networks. The sentence has been modified as follows:
"The three sites are members of the Catalonian air quality monitoring network. Additionally, MSY and MSA are part of the ACTRIS (Aerosol, Clouds and Trace gases Research InfraStructure) and GAW (Global Atmosphere Watch) networks. Aerosol optical properties at the sites are measured following standard network protocols (WMO/GAW, 2016)."

The reference corresponding to the GAW standard protocols is cited in the revised manuscript:
"WMO/GAW report 227: Aerosol Measurement Procedures, Guidelines and Recommendations, 2nd Edition, 2016, 103 pp. August 2016 (WMO-No. 1177)."

**Reviewer#1. Specific comment 6.** P.7 Line 7: "Samples were collected every 3/4 days". I am confused as to whether this means "3/4 day" or "3 to 4 days". I am assuming that the authors imply the former but then remove the 's' from 'days'. This also illustrates the author' usage of past tense to describe the current study. I recommend consistent usage of present tense for this throughout the manuscript.

The frequency of filter sampling was not equally distributed along the whole period. Samples were collected every 3th or 4th day, depending on the sampling period. Therefore, the authors refer to "3 to 4 days"; the expression has been modified accordingly in the revised manuscript.
The past tense used in this sentence makes reference to previous years, when filter samples were collected.

**Reviewer#1. Specific comment 7.** P.7 Line 7: Include a reference for the protocol. There are MANY other instances in the paper where the authors make mention of protocols without providing references.

A reference for the standard gravimetric procedure has been added to the revised paper:
"Alastuey, A., Minguillón, M. C., Pérez, N., Querol, X., Viana, M., and de Leeuw, F.: PM10 measurement methods and correction factors: 2009 status report, ETC/ACM Technical Paper 2011/21, 2011."

**Reviewer#1. Specific comment 8.** P.9 Lines 3-6. "It is remarkable that differences in the sampling conditions (RH, size cut) or chemical analysis methods influence the resulting efficiencies obtained for different emplacements. In this study, scattering RH was controlled below 40% preventing the hygroscopic growth of the particles, which could lead to an enhancement in the scattering efficiency." Please clarify why this is remarkable or else delete or rephrase the sentence. The following sentence is also not true and should be modified or removed. Just because RH<40% does not prohibit scattering enhancements, especially for organics. The water uptake is small (on the order of _10% growth) but not prohibited.
Size cut also influences aerosol intensive properties, including single-scattering albedo, Angstrom exponent, etc. This can be seen from any of the papers based on measurements at the NOAA-GMD monitoring sites (Sheridan et al., 2001; Delene and Ogren, 2002; Sherman et al., 2015; Andrews et al., 2011; : : :..).

Authors agree with the referee. The sentences have been modified accordingly as follows:
"It should be considered that changes in the sampling conditions (i.e. RH or size cut-off) or differences in the chemical analysis methods used on sampled filters can affect the intensive particle optical properties (Delene and Ogren, 2002), and consequently the comparison among the computed MSE and MAE. In fact, the resulting efficiencies can be biased by the cut-off inlet, given

that absorbing aerosols tend to be predominately in the sub-micron fraction (Andrews et al., 2011). In this study both $\sigma_{sp}$ and $\sigma_{ap}$ were collected using a $PM_{10}$ cut off inlet, thus guaranteeing uniformity among the performed optical measurements. An exception occurs at MSA, where a $PM_{2.5}$ cut-off inlet was used until March 2014 and then replaced by a $PM_{10}$ inlet. However, an estimation of the influence of the inlet change on the resulting MSE and MAE at MSA is difficult to achieve, given the relatively short $\sigma_{sp}$ and $\sigma_{ap}$ time series available thus preventing performing two different MLR analyses for the two fractions. Moreover, scattering RH was controlled below 40% at MSY and MSA in order to minimize the hygroscopic growth of the particles and then prevent a significant enhancement in the scattering efficiencies."

**Reviewer#1. Specific comment 9.** **P.10 Line 7: Change 'PM10 levels" to "'PM10 mass concentrations".**

"$PM_{10}$ levels" has been rephrased as "$PM_{10}$ mass concentrations" in all the instances throughout the revised paper.

**Reviewer#1. Specific comment 10.** **P.10 Line 10: Change "PM10 load." To "'PM10 mass concentrations". Please do this for all instances in the manuscript.**

The expression "PM10 load" has been rephrased as "$PM_{10}$ mass concentrations" in all the instances throughout the revised paper.

**Reviewer#1. Specific comment 11.** **Fig.3b. PM10 mass concentration at Montseny is nearly as high in March as in June- July and is higher than in August but this is not discussed at all.**

The following paragraph has been added to the revised manuscript aiming to discuss the aforementioned seasonal variability of PM10 mass concentration:
"It is remarkable the PM10 concentration peak observed in February and March at MSY, which might be attributed to the winter regional pollution episodes typical of the WMB (Pandolfi et al., 2014a). Such scenarios are characterized by anticyclonic conditions which favor the accumulation of pollutants close to the emission sources, and the subsequent transport of pollutants towards the station with the daily increase of the PBL. Pandolfi et al. (2014b) and Pey et al. (2010) reported high nitrate concentrations during these atmospheric conditions at MSY, in agreement with the increased contributions of *Secondary nitrate* shown in Fig. 3b during this time of the year period. The relatively low $PM_{10}$ concentration observed in August at BCN and MSY could be partially explained by reduced anthropogenic activities in the Barcelona metropolitan and industrial areas as a result of the holiday period in Spain. This result is supported by the minima absolute contributions observed in August for *Industrial* and *Traffic* sources at BCN (0.7 and 2.9 μg m$^{-3}$, respectively) and for the *Industrial/Traffic* source at MSY (1.2 μg m$^{-3}$). The higher precipitation rates observed in August compared to June-July

(Perez et al., 2008) could also significantly contribute to reduce $PM_{10}$ concentrations at MSY. Conversely at MSA, the highest $PM_{10}$ concentration was observed in August probably due to the frequent Saharan dust events affecting the mountain top site, in accordance with the highest absolute contribution found for the *Mineral* source in August (3.9 µg m$^{-3}$)."

**Reviewer#1. Specific comment 12. Fig(s).4. Please use site abbreviations as either as sub-plot titles or legend labels in fig(s) 4 to make it easier to look at the figures, without needing to go back and forth between plots and caption to see which plot corresponds to which site.**

The corresponding site acronyms have been added to each plot of Fig. 4.
The correlation plot for the SSA parameter has been removed from Fig. 4

**Reviewer#1. Specific comment 13. P.12 Lines 4-5: The assertion that "Aged organics at MSA (29%) and Ammonium sulfate at MSY (24%) were the dominant sources throughout the year and reached the largest absolute contribution in summer" is not supported by Fig.3. The Mineral source at MSA is comparable to Aged Organics during several months of spring/summer and organics are equal to or exceed ammonium sulfate for several months at MSY. Please reword this assertion to better reflect the data in Fig.3.**

Following the referee's suggestion the paragraph has been modified as follows:
"Differently from BCN, a higher relative contribution of secondary sources, some of them related with natural processes, was observed at MSY and MSA (3b and 3c). Increased contributions of *Secondary sulfate* were observed in summer (29% and 8% at MSY and MSA, respectively), whereas *Secondary nitrate* maximized in winter (17% and 11%). *Aged organics* showed the highest contribution in relative terms in winter (30% and 45% at MSY and MSA, respectively); however the highest absolute contributions were observed in summer (4.8 and 4.1 µg m$^{-3}$). This result is in agreement with the higher SOA formation found at MSA (Ripoll et al., 2015a) and MSY (Minguillón et al., 2015) during the warm period. The *Mineral* source (19% and 27% at MSY and MSA, respectively) maximized in summer, although high contributions were also observed in spring. Similarly to BCN, *Aged marine* (14% and 13% for MSY and MSA, respectively) and *V-Ni bearing* (5% and 11%) sources showed the highest contribution in summer, whereas the *Industrial/Traffic* source maximized in winter (11% and 17%)."

**Reviewer#1. Specific comment 14. P.12 Lines 14-16. This has already been mentioned more than once so the sentence should be deleted.**

The sentence has been removed from the revised paper.

**Reviewer#1. Specific comment 15. P.12 Line 12. Add the word 'Mass' to the beginning of Sect. 3.3 title and to 'absorption efficiencies' and 'scattering efficiencies' throughout this section.**

The text has been modified accordingly to the referee's comment.

**Reviewer#1. Specific comment 16. P.12 Lines 17-24. This passage has already been discussed in previous sections and is not a result. Therefore, it should be deleted.**

The paragraph has been removed from the revised paper.

**Reviewer#1. Specific comment 17. P.13 Line 2: Change the word 'coefficient' to 'MSE'.**

The word "coefficient" has been replaced by "MSE" following the referee's suggestion.

**Reviewer#1. Specific comment 18. P.13 Lines 31-32: "Interestingly, a higher scattering wavelength dependence was observed for those sources with higher contribution from anthropogenic tracers which are mainly present in the fine mode (Table 2)." This is to be expected. Size distributions with higher contributions from the fine mode will possess larger variation of scattering coefficient with wavelength than size distributions with larger contributions from coarse mode aerosol.**

The sentence has been removed from the text.
The results presented for SAE (section 3.3 of the discussion paper) and SSA (section 3.5 of the discussion paper) optical parameters, obtained for specific aerosol sources, have been moved to a new section (Section 3.3.1) included in the revised manuscript.

**Reviewer#1. Specific comment 19. P.14 Line 19: Please clarify what you mean by "European scenarios".**

The paragraph has been modified for a better understanding. Further information on atmospheric European scenarios affecting the MSA mountain-top site can be found in Ripoll et al., 2014 and 2015.

The text in the revised paper is now as follows:
"The large MAE observed for *Secondary nitrate* at MSA ($0.364\pm0.023$ m$^2$ g$^{-1}$) was due to the fact that this source explained around 20% of the measured EC concentration (Fig. 2a). Recently, Ripoll et al. (2015b) have shown the increased concentration of nitrate, ammonium, EC and traffic/industrial tracers at MSA under European scenarios. Such scenarios are characterized by the transport of polluted air masses at high altitude from central and Eastern Europe to the MSA site. This fact may

explain the internal mixing of BC particles in the chemical profile of *Secondary nitrate*, and consequently the high MAE values found for this source at MSA."

**Reviewer#1. Specific comment 20.** P.15 Lines 9-12:" **Both sources presented inverse seasonal cycles following the seasonal variation of mass contributions, with Ammonium sulfate maximizing in summer at MSY (46%) whereas showing similar contribution throughout the year at MSA. Conversely, Ammonium nitrate mainly governed the light scattering in winter (42% and 29% at MSY and MSA)." Please change the wording of the first sentence, as the phrase 'inverse seasonal cycles' is not clear. Wording similar to this is used in other places to describe the cycles and should be fixed. It is clearer to simply state something along the lines of "The annual cycles of ammonium sulfate and ammonium scattering coefficients follow those of the PM10 mass concentration, with summer maxima and winter minima". The assertion that there are similar ammonium sulfate contributions throughout the year at MSA is not supported by Fig.3h, which indicates that the fraction of light scattering attributed to ammonium sulfate is highest in Aug-Sept and lowest in Nov-Dec.**

Following the referee's indications the sentence has been modified as follows:
"The annual cycle of *Secondary sulfate* and *Secondary nitrate* scattering coefficients followed those of the $PM_{10}$ mass concentration, with maxima in summer (46% and 35% at MSY and MSA, respectively) and winter (42% and 29%), respectively."

**Reviewer#1. Specific comment 21.** P.15 Lines 15-17: "Light absorption appeared to be almost dominated by the Traffic source at BCN and in a minor proportion by the equivalent Industrial/Traffic at MSY and MSA (Fig. 3d, e, f), showing high contributions in winter (65%, 42%, 22%) despite the relative low mass concentration (23%, 11%, 17%)." This is one of many instances throughout the paper where the wording probably does not convey the authors' intentions. The meaning of 'almost dominated' is unclear. Light absorption coefficient is either dominated by Industrial/Traffic or it is not. Based on Figs.3d-3e, it looks as if it is only dominated by Industrial/Traffic during non-summer months at BCN, although it**
**clearly exerts the major influence (_40%) at MSY. I also do not understand the meaning of "a minor proportion by the equivalent Industrial/Traffic at MSY and MSA". The influence of Industrial/Traffic (_40-50% at MSY and _20% at MSA) is neither dominating or minor. Please clarify this and similar statements throughout the document so that their meaning reflects your intentions. You do a much better job on P.15 Lines 30-31.**

Following the referee's suggestion the sentence has been modified as follows:
"The *Traffic* source at BCN and the *Industrial/Traffic* source at MSY clearly exerted the major influence on light absorption contributing 54% and 41% to $\sigma_{ap}$, respectively, despite the relative low $PM_{10}$ contributions (16% and 10%, respectively). Maxima contributions were observed in winter at

BCN for the *Traffic* source (65%) and in October-January at MSY for the *Industrial/Traffic* source (46%), however a lower influence of *Industrial/Traffic* was observed on average at MSA (18%) (Fig. 3 d, e, f)."

"P.15 lines 30-31" have been removed from the revised paper.

**Reviewer#1. Specific comment 22. P.15 Lines 20-21: "Therefore, Traffic, Industrial/Traffic and V-Ni sources which highly influence air quality also have caused an important effect on radiative forcing, particularly in those sites closer to the emission sources." This assertion may be true but cannot be supported, given the measurements available in the current study. Only absorption coefficient data is available at BCN (the main site influenced by these sources) and radiative forcing depends primarily on aerosol optical depth (which in turn is largely controlled by scattering coefficient). As such, I recommend that the authors either remove this statement or provide more support for it, given the available data.**

The authors agree with the referee, this study does not include any calculation on radiative forcing. The term "radiative forcing" has been removed from the revised paper.

The sentence has been modified as follows:
"Therefore, *Traffic*, *Industrial/Traffic* and *V-Ni bearing* sources, which highly influenced air quality, also significantly contributed to $\sigma_{ap}$, and especially in those sites closer to the emission sources."

**Reviewer#1. Specific comment 23. P.15 Line 26-P.16 Line 3: A summary of results should be placed in the Summary and Conclusions section, not in the Results section.**

According to the referee's comment the paragraph has been removed from the paper.

**Reviewer#1. Specific comment 24. P.15 Lines 26-27: "As a summary, we have shown that the main target pollutant sources affecting air quality degradation have caused important effects on light extinction in the northwestern Mediterranean." This sentence should be clarified. You cannot state anything regarding light extinction at BCN without measurements of light scattering coefficient, which represents the major contribution to light extinction coefficient.**

The authors agree with the referee, the sentence has been removed from the paper.

**Reviewer#1. Specific comment 25. P.16 Line 1: Please remove the phrase "As a novelty,".**

The expression "As a novelty" has been removed from the text.

**Reviewer#1. Specific comment 26.** P.16 Line 19: What 'published results'? Please cite reference(s) to support this.

The sentence has been modified as follows:
"According to published results (Ryan et al., 2005 and references therein)"

**Reviewer#1. Specific comment 27.** P.17 Line 4: State at which site you are discussing the SSA. You state that it is MSY in the caption of Fig.4 but it should also be stated in your discussion, so that the reader does not need to go back and forth between the discussion and figure captions.

According to the referee's general comment 2 and specific comment 30, the results presented for the simulation and trend analysis of the SSA parameter have been removed from the revised paper.

**Reviewer#1. Specific comment 28.** P.17 Line 5: The slope of modeled versus measured SSA is 0.80, which is not close to 1. Please reword this.

The sentence is not included in the revised manuscript given that the simulation of the SSA parameter has been removed from the text.

**Reviewer#1. Specific comment 29.** P.17 Line 15: Since trend study was only done for MSY, the section title should be rephrased to more accurately represent the section contents. This could be as simple as "Long-term trends in scattering and absorption coefficients at MSY".

The title of section 3.6 has been rephrased accordingly to the referee's suggestion.

**Reviewer#1. Specific comment 30.** P.17 Lines 19-21: "Despite a larger uncertainty was found for the modeled SSA, this technique allowed to further investigate the temporal trend of this important parameter and its relation with changes in atmospheric composition (Fig. 7)." Marginal agreement of measured and model SSA (R2=0.42, slope =0.80) during the 2010-2014 period at MSY (when agreement should be best, since the authors use the same period to both generate and evaluate the model) likely makes it impossible to state much about any long-term trends in SSA at MSY. As such, I believe that the discussion of SSA trends at MSY should be removed from the paper.

Authors agree with the referee's comment, the low correlation obtained for the modeled-observed pairs of SSA prevent to accurately estimate any trend for the SSA parameter. Following the referee's suggestions the SSA trend study at MSY has been removed from the revised manuscript.

**Reviewer#1. Specific comment 31. P. 18 Lines 4:6:" A marked decline was also observed for nitrate and sulfate PM in other European monitoring sites since 1990, as outlined in the EMEP report 1/2016.". Please provide a reference for the EMEP report. Please do the same for other instances where sources are named but not referenced.**

The following reference has been included for the EMEP report:
"as outlined in the EMEP report 1/2016 (Colette et al., 2016)."
"Colette et al., 2016. Air pollution trends in the EMEP region between 1990 and 2012. Joint Report of the EMEP Task Force on Measurements and Modelling (TFMM), Chemical Coordinating Centre (CCC), Meteorological Synthesizing Centre-East (MSC-E), Meteorological Synthesizing Centre-West (MSC-W). Kjeller, NILU (EMEP: TFMM/CCC/MSC-E/MSC-W Trend Report) (EMEP/CCC, 01/2016)."

**Reviewer#1. Specific comment 32. P.18 Lines 7-17: Please cite references for all of the claims made in this paragraph, since they are not supported by the data presented in this manuscript. Were the claims from the Pandolfi et al. (2016) paper? If so, you should state this and probably not spend a paragraph summarizing a different study. Simply state (as you did in first sentence of the paragraph) that the causes of reductions in most sources at MSY are discussed in Pandolfi and then move on to how your study contributes to the knowledge. If not, please provide the sources to justify the assertions.**

Authors consider that the causes leading to a reduction in some of the chemical compounds and sources at MSY should be included in the text, in order to better assess the scattering and absorption optical trends and its relation with trends in atmospheric composition.
Assertions regarding decreasing trends for the chemical compounds and sources at MSY are based on the results published by Querol et al., 2014 and Pandolfi et al., 2016. These references have been included throughout the paragraph.

The text in the revised paper is now as follows:
"Querol et al. (2014) and Pandolfi et al. (2016) investigated trends of PM chemical components and aerosol sources at MSY, providing further explanation on the causes leading to the reduction of the atmospheric pollutants in the area. The financial crisis affecting Spain from 2008 contributed to reduce the ambient PM concentrations. A decrease in *Secondary nitrate* can be explained by the reduction of ambient $NO_X$ and $NH_3$ concentrations (Querol et al., 2014). The decreasing trend of the *Secondary sulfate* source may be supported by the reduction of sulfate particles, mainly attributed to the gas desulfurization at several facilities (Pandolfi et al, 2016). A decrease in secondary sulfate may be also explained by the 75% reduction of $SO_2$ concentration in the Barcelona harbor, supported by the regulation of sulfur content in shipping emissions in EU harbors from 2010 (Schembari et al., 2012). This regulation together with the 2007 ban around Barcelona on the use of heavy oils and petroleum coke for power generation, which contributed to a drastic decrease in V

and Ni concentrations (Querol et al., 2014), were the main reasons supporting the observed reduction of the contribution of the *V-Ni bearing* source. "

**Reviewer#1. Specific comment 33.** P.18 Lines 22-23: **"Interestingly, the SSA showed a significant decreasing trend of -0.11 % y-1 (-0.001 yr-1) leading to a total reduction (TR) of 1.24 % since 2004 at MSY, pointing that the atmosphere is getting significantly darker." A change in SSA of 0.01 is relatively small and does not imply that the atmosphere is getting significantly darker.**

Authors agree with the referee.
The sentence is not included in the revised manuscript given that the SSA trend study has been removed from the paper.

**Reviewer#1. Specific comment 34.** P. 18 Lines 25-28: **"Differences in the SSA reduction at both sites might be explained by the severe pollution episodes taking place in the Po Valley resulting in a higher dominance of absorption in the light extinction process, compared to MSY which is representative of a less polluted environment." You should provide some reference for this. If not, the decreasing SSA could just as easily be due to larger reductions in scattering than absorption.**

These lines refer to the study published by Putaud et al. (2016). This study showed that the increasing biomass burning emission sources during the colder period contributed to decrease the SSA parameter in the Po Valley. In order to support this result, Putaud et al. (2016) showed that the ratio absorption coefficient/EC concentration increased between 2005 and 2010, especially for the highest values which were observed during the colder months. This result was attributed to the increasing concentrations of other light-absorbing substances, such as brown carbon, during cold months, when wood burning for domestic heating was used more and more in northern Italy.
The reference "Putaud et al. (2016)" has been added in more than one instance throughout the paragraph.

Given that the SSA trend study was removed from the text the paragraph has been rephrased as follows:
"Statistically significant downward trends of PM mass concentration, $\sigma_{sp}$, $\sigma_{ap}$ and SSA were found in the Po valley (Italy) for the period 2004-2010 (Putaud et al., 2014). A higher decreasing rate was observed for $\sigma_{sp}$ (-2.8 % yr$^{-1}$) compared to $\sigma_{ap}$ (-1.1 % yr$^{-1}$), likely due to the increasing contribution of light-absorbing organic matter to light absorption during cold months in the Po Valley (Putaud et al., 2014). In the present study, smaller differences between $\sigma_{sp}$ and $\sigma_{ap}$ were observed at MSY, accounting the total reduction trends for -50% and -45%, respectively. This fact might be explained by the different background sites considered; whereas the Po Valley is a highly polluted area, MSY is representative of a cleaner environment where biomass burning emissions, which highly contribute to light absorption, are considerably lower (Minguillón et al., 2015; Ealo et al., 2016)."

**Reviewer#1. Specific comment 35.** P.18 Lines 29-34: This is another example of claims being made (and percentages brought up) without any references for these numbers or results. Please include sources/references for these numbers.

The paragraph has been removed from the text.
Most of section 3.6 has been rephrased and changes are directly included in the revised paper.

**Reviewer#1. Specific comment 36.** P.19 Lines 12-13: "The reduction in the SSA trend points out the increasing prominence of absorption in the light extinction process in the NW Mediterranean." This represents another claim that is unsupported by the data. An apparent 1% decrease in SSA at a single site does NOT point to increasing prominence of absorption in the light extinction process in the NW Mediterranean region. Please either provide more support for this assertion or delete it.

Given that the SSA trend study was removed from the paper, section 3.6 has been substantially modified. Changes are directly shown in the revised paper.

**Reviewer#1. Specific comment 37.** P. 19 Lines 16-23: This discussion of radiative forcing and relation to policy lies well beyond what can be stated, based on data/results from this manuscript. Please either change the wording so that it is supported by your results or else remove.

Authors agree with the referee. The term "radiative forcing" and the conclusions based on this approach have been removed from the paper. Given that SSA trend analysis was removed from the text, section 3.6 and conclusions of the revised manuscript have been substantially modified. Changes are directly shown in the revised paper.

**Reviewer#1. Specific comment 38.** P. 20 Lines 31-32: "A total reduction (TR) of -1.12% in the SSA was mainly motivated by the heterogeneous and non-selective reduction of key aerosol sources showing opposite effects on radiative forcing." The assertion regarding effects on radiative forcing is not supported by the data and should be removed.

We agree with the referee, the sentence has been removed from the text. Following previous referee's suggestions the term radiative forcing has been removed from the paper.

**Reviewer#1. Specific comment 39.** P.20 Line 34-Page 21 Line 8: "However, these measures have resulted in a more pronounced reduction of light-scattering aerosol sources (Ammonium nitrate, Ammonium sulfate), leading to an increase of the incoming solar radiation and therefore

**contributing to climate warming. This positive radiative effect is enhanced by the less effectiveness of air quality strategies for reducing light-absorbing sources containing dark particles. A decrease in the SSA trend points to a darkening of the atmosphere and consequently to a progressive predominance of absorption in the light extinction process in the NW Mediterranean. Accordingly to the results presented in this work, future strategies need to focus on preferentially reducing atmospheric aerosols mainly originated from combustion sources. Industrial/Traffic and V-Ni aerosol sources, which highly contributed to air quality degradation but also to light absorption, should be abated thus addressing win-win policies aimed to improve air quality and mitigate climate warming**
**in the NW Mediterranean."**

**Where is the evidence for increasing in solar radiative and contributions to climate warming? This and the policy recommendations are completely unsubstantiated by the data and either need to be justified or removed. A small and questionable decrease in SSA (based on poor correlation and marginal agreement between modeled and measured SSA during common period) at a single site cannot be used to make claims regarding darkening of the atmosphere. See my many comments above regarding assertions that are unsupported.**

Authors agree with the referee, we do not provide any results on radiative forcing, nor solar radiation measurements, for this reason the term "radiative forcing" and the conclusions making reference to this term have been removed from the paper.

Moreover, as outlined in previous comments, the SSA trend analysis has been removed from the manuscript and therefore section 3.6 and section 4 have been substantially modified in the revised paper. In the following lines it is shown the last paragraph from section 3.6 of the revised paper, as a representation of the modifications carried out in this section. Additional changes introduced in the manuscript are directly shown in the revised paper.

"Further research on light scattering and absorption long-term trends and its relation with changes in atmospheric composition is needed to better understand the role of aerosols on optical properties and on the climate system. Based on the published studies and the present results, further efforts focusing on the reduction of atmospheric pollutants containing BC particles (mainly emitted from fossil fuel combustion and biomass burning sources) need to be addressed. Given the toxicity of their chemical tracers, as well as their large contribution to light absorption, *Industrial/Traffic* and *V-Ni bearing* sources must be reduced through the implementation of win-win policies, aiming to improve air quality and public health, and mitigate climate warming."

**Anonymous Referee# 2.**

**This study showed not only aerosol source contributions based on a positive matrix factorization (PMF) analysis, but also mass scattering and absorption efficiencies (MSE and MAE) of different**

aerosol sources by using multilinear regression method at urban, regional and remote backgrounds in the Spain. Although the results and discussion were documented well, there are several important shortcomings.

**Reviewer#2. Specific comment 1.** While the detailed pedagogical description of the approach is appreciated, the paper overall must be shortened. There are many repetitions in the text or not essentially needed with many references. This review strongly suggest that text must be made more succinct. For example, in INTRODUCTION, the authors explained overall aspects of aerosols in climate-air quality research. Most of them are overstated and not directly related with the results of this study. The words "From air quality to climate" in title also should be removed. Summary and conclusions also should be shortened; highlight the major findings succinctly and provide directions/implications of this work.

Following recommendations from both referees, some sections of the manuscript including abstract, introduction and conclusions have been rephrased in the revised manuscript. Those parts of the text which were repeatedly commented have also been removed or rephrased. Overall, substantial changes have been introduced in the revised paper aiming to improve readability of the text and better reflect the achieved results.
The title has been modified accordingly to the referee's comment.

**Reviewer#2. Specific comment 2.** Generally, source apportionment by PMF model shows the contributions of aerosol chemical properties from various sources, such as traffic (vehicle), biomass burning, dust (road dust), marine, industry, secondary nitrate, secondary sulfate, ship, etc. However, the sources given in this study are aerosol compositions, except for industrial/Traffic and Marine. Firstly, the authors should show the chemical compositions and discuss the characteristics during the study period. The explanation of major emissions sources of the aerosols also provided. Secondly, more detailed descriptions for source profiles should be given. For example, V-Ni at MSY originated mainly form shipping emissions (see section 3.3), why EC is not considered at the source profile.

According to the Referee's suggestion, the name of *Ammonium nitrate* and *Ammonium sulfate* aerosol sources have been replaced by *Secondary nitrate* and *Secondary sulfate*, respectively.

The aerosol sources and contributions discussed in the present paper for BCN and MSY were firstly published by Pandolfi et al. (2016) for the same study period. Therefore, further information on the results of PM10 and PM2.5 chemical compounds and PMF source contributions at BCN and MSY can be found in Pandolfi et al. (2016). Authors consider that an extended discussion on the PMF results at BCN and MSY would provide repetitive information.

The novelty of the present paper, regarding PMF analysis, are the results presented for the source chemical profiles and contributions at MSA, and that's the reason why results from PMF at MSA are extensively discussed in the paper compared to the results presented for MSY and BCN, which were firstly and widely discussed in Pandolfi et al., 2016.

However, the resulting PM source contributions are just a tool to obtain the main goal of the present paper which is focused on providing, for the first time, mass scattering and absorption efficiencies (MSE and MAE) for specific aerosol sources. Additionally, the paper aims to assess the PMF-MLR technique to accurately simulate scattering and absorption coefficients when chemical speciation data is available.

A discussion on the PM chemical compounds at MSA is not provided given that doing so detracts the focus of the paper. Previous studies deployed at MSA have presented the PM10 and PM1 chemical composition at this site (Ripoll et al., 2015).

Authors agree with the referee, further information on the source chemical profiles at BCN and MSY should be provided in the paper. The source chemical profiles for BCN and MSY have been added to the revised supporting material. Even though this information was previously reported in Pandolfi et al. (2016), it is needed to better understand the changing chemical profile of the common aerosol sources at the three different sites.

The following Figure S2 has been added to the revised supporting material.

[Figure]

Figure S2. Source chemical profiles obtained by means of the PMF model at BCN and MSY for the period 2004-2014 (Pandolfi et al., 2016).

We guess that the referee is making reference to MSA in the last sentence, given that the unique chemical profile resulting from applying a PMF model shown in the discussion paper makes reference to the results presented for MSA.

It should be noted that the PMF model was run automatically and then we cannot choose the apportionment of specific species to the different sources, so that the *V-Ni bearing* source at MSA, constituted by specific chemical species, is an automatic output from the PMF model. Differently from BCN and MSY, the resulting *V-Ni bearing* source at MSA does not include EC in the chemical profile given that EC is mainly attributed to the *Aged organics* and *Ammonium nitrate* sources at this site. This fact might be related to the high altitude and far position of MSA from the Mediterranean coast and shipping emissions. Whereas MSY is affected by the direct shipping emissions from the Mediterranean coast, the MSA site might be influenced by atmospheric long range transport, and therefore the *V-Ni bearing* source at both sites might be internally mixed with different chemical species.

The following sentence has been added to the revised manuscript:

"Differently from BCN and MSY, the *V-Ni bearing* source at MSA was not enriched in EC possibly because of the high altitude of this station and its position, far from the western Mediterranean coastline and shipping emissions."

**Reviewer#2. Specific comment 3.** **It is not fully explained how the authors determined the scattering and extinction efficiencies (MSE and MEE) of chemical species by using the Multilinear regression method. Eqs. 1 and 2 are not enough. This reviewer strongly suggests that detailed steps for getting the MSE and MEE of chemical species by using the multilinear regression have to be present in main text or in the supplement.**

The MLR analysis for estimating MSE and MAE of aerosol particle sources is based on the IMPROVE algorithm (Hand and Malm, 2007 and references therein). Equations 1 and 2 presented in section 2.3 of the discussion paper are similar to those equations showed in previous published studies. Authors do not exactly understand which changes should be introduced in the MLR equation.

Section 2.3 (named section 2.4 in the revised paper) has been rephrased for a better understanding. Changes have been directly modified in the revised paper.

**Reviewer#2. Specific comment 4.** **Regarding to the reconstruction (section 3.5), what is the root-mean-square difference for data given in Figure 4? This reviewer suggests to use percentage difference of bias rather than fractional bias.**

Following the referee's comment, the root-mean-square error (RMSE) has been accordingly calculated for the modeled and measured optical parameters. The RMSE is widely used in modeling evaluation and provides valuable information on the accuracy of the predicted coefficients compared to the observational data.

The following paragraph has been added to the revised manuscript:
"The root mean square error (RMSE) was calculated for the observed-modeled datasets, showing low dispersion and high accuracy in the modeled values. Scattering and absorption coefficients were well reproduced by the model, showing RMSE values of 8.76 and 6.06 $Mm^{-1}$ for $\sigma_{sp}$ at MSY and MSA, and values of 2.61, 0.55 and 0.23 $Mm^{-1}$ for $\sigma_{ap}$ at BCN, MSY and MSA, respectively."

The Fractional bias (FB) is presented in this study, given that published studies applying a MLR model for MSE calculation have also used the FB parameter to assess the accuracy of the model to simulate $\sigma_{sp}$ coefficients (Ryan et al., 2005). Therefore, FB has also been used in this study to compare our results with the published studies. The replacement of fractional bias by the percentage of FB does not provide further information on the discussion of the accuracy of the modelled $\sigma_{sp}$ coefficients. Authors prefer to keep the FB parameter in the revised paper.

**Reviewer#2. Specific comment 5.** **Long-term trends in Section 3.6: There is no detailed explanation on Figure 6. There is a critical issue that how we can trust these reconstructed values given in Figure 6 and associated trend analysis results. For example, Collaud Coen et al (http://www.atmos-chem-phys.net/13/869/2013/acp-13-869-2013.pdf) reported that, in the Europe, significant trends were not observed for aerosol optical properties at most sites (see Table 8 in the paper). However, this study, this study reports about a scattering TR (total reduction) of -52% since 2004 and -44% in the absorption coefficient. Therefore, the authors should investigated all published literature, especially in Europe, and give a reasonable and reliable explanation with evidences. If there are such distinct decreases in aerosol scattering and absorption properties, is this trend also consistently seen from aerosol optical depth measurements from AERONET/SKYNET and satellite sensors? Also, do PM2.5 concentrations show a decreasing trend over the study period?**

A very good agreement has been found between measured and modeled pairs of scattering ($R^2$=0.88) and absorption ($R^2$=0.8) with slopes close to one at MSY for the period 2010-2014 (Fig. 4 of the paper). In addition, following the suggestion of referee#1 in general comment 1, an independent subset of the period (year 2015) not considered in the MLR analysis has been used in order to further evaluate the accuracy of the modeled scattering and absorption (see general comment 1). Good agreement was found between modeled and measured $\sigma_{sp}$ (R2=0.85) and $\sigma_{ap}$ (R2=0.76) coefficients, at 525 and 637 nm respectively, showing slopes close to one for the year 2015 at MSY (Fig. 6 of the revised paper). Authors consider that these results support the accuracy of the modeled $\sigma_{sp}$ and $\sigma_{ap}$ time series in order to carry out the trends study.

Pandolfi et al. (2016) observed significant decreasing trends in PM10 and PM2.5 chemical species concentration and PMF sources contribution for the period 2004-2014 at the BCN and MSY sites. The reference of Pandolfi et al. (2016) is cited throughout the paper, given that PMF source contributions

at BCN and MSY are also used in this study, as is detailed in the discussion paper. Therefore, these results showing decreasing trends on PM10 and M2.5 concentrations and their chemical tracers support the observed reduction on the optical properties trends at both sites.

Other studies have been published in the last years showing clearly that the concentrations of PM and other air pollutants such as sulfur dioxide (SO2) and carbon monoxide (CO), have markedly decreased during the last 15 years in many European countries (EEA, 2013; Barmpadimos et al., 2012; Cusack et al., 2012; Querol et al., 2014; Guerreiro et al., 2014 among others). Cusack et al. (2012) reported the reduction in PM2.5 concentrations observed at regional background (RB) stations in Spain and across Europe. Barmpadimos et al. (2012) have also shown that PM10 concentrations decreased at some urban and rural background stations in five European countries. Henschel et al. (2013) reported the decrease in SO2 levels in six European cities. EEA (2013) also reported general decreases in NO2 concentrations.

A less number of studies focused on optical properties trends analysis were published in Europe. For example, Putaud et al. (2014) found downward trends for PM, scattering, absorption and SSA series for the period 2004-2010 in the northeast of Italy.

Few studies clearly support the reduction of the atmospheric PM in many European measurement sites, which consequently leads to a decrease in the scattering and/or absorption trends.

Aerosol optical properties measurements are relatively recent in Europe compared to the EEUU, as shown in Collaud Cohen et al., 2010. The latter study presents a global overview on trends study of optical parameters, however no significant decreasing trends were observed in the European sites. This fact may be due to differences in the time period for major European reductions relative to the time period covered by the aerosol optical property measurements. The limited number of available datasets as well as the characteristics of the sites, which might not be representative for trends study (e.g. high altitude or marine), also supposed a limiting factor for trends study (Collaud Cohen et al., 2010).

Columnar measurements are affected by atmospheric processes and emission sources at regional and continental scales, involving transboundary transport. However, surface data is mainly affected by local and regional emission sources which are directly influenced by the air quality regulation strategies. Authors consider that aerosol optical depth measurements should not be included in the revised paper given that the paper focuses on surface measurements, and the decreasing rates of columnar optical parameters cannot be explained by the process occurring at the surface. The trends study of surface scattering and absorption series aims to relate the trends of specific aerosol sources with trends in optical parameters, in order to assess the effects that air quality strategies adopted in recent years are having on light extinction. Therefore, authors think that a trend study of columnar measurements does not fit in the present paper; however it is a good suggestion that will be considered for future studies.

[revised manuscript text omitted]

quantitative statistics. With this aim, the root mean square error (RMSE) and fractional bias (FB) described in equation 3 (Ryan et al., 2005), were computed for modelling evaluation. FB is described in Eq. 3 (Ryan et al., 2005),, Wwhere $\sigma_{sp}Se^{sim}$ is the modeled scattering coefficient and $\sigma_{sp}Se$ is the measured value, for each daily data point.

$$FB = \frac{\sigma sp Se^{sim} - \sigma sp Se}{\sigma sp Se}$$
(Equation 3)

A total of 303, 379 and 503 daily data points were used in the MLR analysis for source apportionmenting analysis ofto absorption at MSA, MSY and BCN, respectively, . Wwhereas 222 and 307 daily data points were considered for MSE calculation at MSA and MSY. , ensuring that the number of samples is large enough to provide stable results.

**2.56 Statistical tests for trends study**

The Theil-sen slope estimate (TS) (Theil 1950; Sen 1968) is a non-parametric test which was investigated for the monthly averages of light scattering and, absorption and SSA in order to test for the occurrence of a non-null slope in the data series during the period 2004-2014 at MSY. The total and annual reduction of these optical parameters was investigated using bootstrap resambling for the monthly deseasonalized time series, reducing the possible influence of outliers on trend estimates and obtaining robust slope p-values.

A multi-exponential fit developed within the Task Force on Measurements and Modelling (TFMM) by the Meteorological Synthesizing Centre – East (MSC-E) group (Shatalov et al., 2015), aiminged to studyat studying temporal trends of air pollution in the multi-exponential form (Shatalov et al., 2015), was used for representing the decomposed modeled monthly temporal series in: main component, seasonal component and residual component. Additionally, this technique allowed us to estimate the non-linearity (NL) parameter for the trends. An NL of 10% was used as threshold to define a linear trend (NL<10%).

**3. Results**

**3.1 Source profiles and contributions to PM$_{10}$**

Seven aerosol sources were identified at MSA in the PM$_{10}$ fraction by performing a PMF model analysis for at the MSA station during the period 2010-2014. The source chemical profiles and source contributions to the measured PM$_{10}$ mass are shown in Fig. 2 and Table 1. These results will be studied together with the Average absolute and relative chemical profiles (Fig. S2 ) and source contributions (Table 1) obtained for previously quantified by Pandolfi et al. (2016) at BCN and MSY forduring the period 2004-2014., also 
[revised manuscript text omitted]
 intenseive sea breezes circulations transporting pollutants to inland regions.; Aaverage contributions to $\sigma_{ap}$ during this seasonin summer were 31% at BCN, 17% at MSY and and around 167% at MSY and MSA. Therefore, *Traffic*, *Industrial/Traffic* and *V-Ni bearing* sources, which highly influenced air quality, also have caused also significantly contributed to $\sigma_{ap}$. important effect on radiative forcing, particularly and especially in those sites closer to the emission sources. *Aged organics* became a relevant source in the absorption process at the regional and remotecontinental background sites contributing (20% and 32%), respectively, due to both its large contribution toin PM$_{10}$ and its relatively large MAE compared to other sources. *AmmoniumSecondary* sulfate contributed on average by 10%, 16% 
[revised manuscript text omitted]

[Figure]

[Figure]

**Figure 1**

[Figure]

5 **Figure 2**

[Figure]

**Figure 3**

[Figure]

**Figure 4**

[Figure]

**Figure 5**

[Figure]

**Figure 6**

a)

[Figure]

a) BCN

[Figure]

b)

[Figure]

b) MSY

[Figure]

c)

[Figure]

c) MSA

[Figure]

**Figure 76**

[Figure]

**Figure 7**

[Figure]

**Figure 8**

[Figure]

Figure 8

**Supporting material**

[Figure]

[Figure]

**Figure S1.** Correlation matrix between pairs of aerosol sources obtained by means of the PMF model at a) BCN, b) MSY and c) MSA. The correlation is coded by shape and colour for better visualization, lower correlations are represented by circles and lighter colours whereas higher correlations are represented by ellipses and darker colours.

[Figure]

**Figure S2.** Source chemical profiles obtained by means of the PMF model at MSY and BCN for the period 2004-2014 (Pandolfi et al., 2016).

---

## Author Response (AR2)

**Reply to reviewers**

**ACP-2017-217**

**Impact of aerosol particle sources on optical properties at urban, regional and remote levels in the north-western Mediterranean**

The authors would like to thank the reviewers for their comments and suggestions, which helped improving the quality of this work. A new version of the manuscript has been prepared following the referee's suggestions. We provide below detailed replies to each of the comments.

Anonymous referee#3

This study presents sources of PM10 and their MSEs /MAEs in at an urban (Barcelona-BCN), a regional (Montseny-MSY) and a remote (Montsec-MSA) background sites in the northwestern (NW) Mediterranean. The authors have applied PMF model to investigate the aerosol MSEs /MAEs from PM10 sources. However, I also feel that there are two critical issues that need to be addressed before the next step of ACPD.

**Reviewer#3. Specific comment 1. According to Mie theory, aerosol MSE is determined by aerosol size and density (or chemical species). Also, aerosol MAE is determined by the core sizes of EC, the coating shell size of non-EC absorption matters (e.g. organic matters or brown carbon), and mixing states. As a result, aerosol MSE should be constant when aerosol size and density were known, no matter what the internal or external mixing were. Thus, I don't think the summary in line 5-10 of page 4 is correct.**

We agree with the Referee, there is a mistake in the last sentence of the paragraph, the variability of MSE is not directly related with the mixing state but is dependent on particle size and density (Hand and Malm, 2007). The sentence indicated by the referee has been removed from the text, and the paragraph is now as follows:

"However, none of the published studies dealing with the estimation of MSE have considered the internal mixing state of atmospheric aerosols, given that each chemical specie was treated separately from the other."

These two sentences are shown in the paper aiming to explain the MSE variability as a result of the changing aerosol size and density:

"The mass scattering and absorption efficiencies (MSE and MAE, respectively) are key intensive optical parameters that relate the mass concentration of specific chemical species to the particle light scattering ($\sigma_{sp}$) and absorption ($\sigma_{ap}$) coefficients. These intensive optical parameters depend on

intrinsic aerosol properties, such as particle effective radius, particle mass density or refractive index, and they are very useful to better parameterize the aerosols direct radiative effect in atmospheric climate models (Seinfeld and Pandis, 1998)."

"MSE for Secondary sulfate were quite different between MSY and MSA (4.5±0.2 and 10.7±0.5 m$^2$ g$^{-1}$, respectively), probably due to differences in the source origin and the related particle size."

The MAE is usually estimated for EC, and its variability can be explained by the mixing state and the possible coating with non-absorbing material. However, it is important to highlight that we are not presenting the absorption efficiency of EC. Our study aims to present the MAE of mixed particles emitted by different sources, given that EC concentration is explained by the different sources presented in the paper. As a result all the PMF sources contributed to light absorption showing a feasible MAE value, differently from previous published studies where MAE value is entirely attributed to EC (and OC in some studies). Even those sources usually treated as pure scattering particles in climate models (such as secondary sulphate and secondary nitrate) also showed a certain degree of absorption efficiency due to the internal mixing with EC, as shown in the source chemical profiles (Fig.2 and Fig. S2). Therefore, the MAE variability among different PMF aerosol sources at the three sites can be attributed to the internal mixing of chemical species constituting the source.

**Reviewer#3. Specific comment 2. The measured dry scattering coefficients (σsp) could be underestimated due to the loss of volatile chemical species (e.g. nitrate, semi-volatile organic compound) by heater in nephelometer (Bergin et al., 1997, Environ. Sci. Technol. 1997, 31, 2878-2883). However, aerosol chemical compositions concentrations were determined from filter samples which collected under the ambient condition. The estimated MSE from PM sources should be thus overestimated. It should be also pointed out that the MSEs of nitrate and sulfate far exceed their theory values.**

Authors agree with the referee, the measured dry scattering coefficients could be underestimated due to the loss of volatile chemical compounds when the nephelometer cell is heated. This fact is a common process in all previous studies presenting MSE calculated from nephelometer measurements, and may results in an underestimation of the MSE.

However, it is also known the volatilization of low volatile aerosol components collected by filter sampling at ambient conditions. For example, Minguillón et al. (2015) showed that nitrate and ammonium concentrations determined by ACSM instrument and filter sampling at MSY site differs more than 20% for the PM1 fraction. Therefore, it is clear that the volatilization of chemical compounds is reflected in both variables, scattering measurements and chemical species concentration.

Moreover, the average temperature (±SD) registered inside of the nephelometer cell (22.6±4.6°C) and inside of the sampling station (equivalent to the filter temperature conditioning; 23.7±4.5°C) shows similar values for the period 2011-2015. It is remarkable that 5$^{th}$ and 95$^{th}$ percentiles of the

nephelometer cell temperature are 16±3.5 and 29.9±2.2, respectively, verifying that most of the temperature data points are close to the mean value.

As a result, an underestimation of the scattering measurements may be compensated with the volatilization of some chemical compounds collected on sampled filters and the consequent underestimation of mass concentration, given the similarity of temperature observed for the nephelometer cell and the filter conditioning.

The high MSE obtained for the Secondary sulfate source at MSY is in agreement with the result showed by Pandolfi et al. (2014) at the same site, where even a higher MSE of 15.6±2.8 $m^2\,g^{-1}$ was observed for sulphate chemical compound.

**Anonymous Referee#4**

**In this paper, the authors investigate the mass scattering and absorption efficiencies (MSE and MAE) of different aerosol particles sources at urban (BCN), regional (MSY) and remote (MSA) backgrounds in the Northwestern Mediterranean using PM10 chemical speciation and particles optical properties. The authors propose a new approach aiming to apportion the PM10 source contributions, identified by means of the Positive Matrix Factorization (PMF), to the measured particle σsp and σap coefficients. This approach has the advantage that the computed MSE and MAE take into account the internal mixing of atmospheric particles. The paper is interesting for ACP, and the introduction, methodology, discussion and results are well documented, but requires corrections.**

**Reviewer#4. Specific comment 1. The authors use acronyms in text that are not defined previously. As example: P.3 Line 20 (NWM), P.5 Line 10 (WMB), P.10 Line 10. Also, sometime the authors use in text the acronyms and sometimes the words of these acronyms. Please, check the text carefully.**

Acronyms in the text have been checked and corrected in the revised paper. NWM is now cited in the text as NW Mediterranean, and WMB makes reference to the western Mediterranean Basin.

**Reviewer#4. Specific comment 2. P.6, 2.2 Section. Please, include a Table containing sampling sites, measurement period, measured parameter, … there is confusion in text and it's not easy to follow the measured variables in each measurement station, as well as the measurement period. As example: P.6, Line 4, measurement period from MSY. P.6, Line 13, measurement period at BCN, MSY and MSA from aerosol absorption coefficient. The same from gravimetric PM10 mass concentrations. P.6, Line 5: "σsp measurements at BCN are not available", but in Line 31: "Optical measurements were considered for the periods 2009-14 at BCN…". Please, clarify.**

The text has been modified accordingly to the referee's recommendation. Sampling period at the three sites has been commented in each of the paragraphs of section 2.2, when the different measured parameters are presented, instead of doing it at the end of section 2.2.

The following sentences have been modified as follows in the revised paper:

"$\sigma_{sp}$ measurements were collected at MSY for the period 2010-2014 using a $PM_{10}$ cut-off inlet. Measurements at MSA were carried out using a $PM_{2.5}$ cut-off inlet from 2011 until March 2014, and then replaced with a $PM_{10}$ cut-off inlet."

"$\sigma_{ap}$ measurements were collected at BCN, MSY and MSA for the periods 2009-2014, 2010-2014 and 2011-2014, respectively."

"Samples were collected every 3 to 4 days on 150 mm quartz micro-fiber filters (Pallflex 2500 QAT-UP and Whatman QMH) using high-volume samplers (DIGITEL DH80 and/or MCV CAV-A/MSb at 30 $m^3$ $h^{-1}$) for the periods 2004-2014 at BCN and MSY, and for the period 2010-2014 at MSA."

The following table has been added to the revised supplementary material:

| Parameter | BCN | MSY | MSA |
|---|---|---|---|
| $\sigma_{sp}$ | - | 2010-2014 | 2011-2014 |
| $\sigma_{ap}$ | 2009-2014 | 2010-2014 | 2011-2014 |
| Species concentration | 2004-2014 | 2004-2014 | 2010-2014 |

Table S1. Study period considered for the measured parameters ($\sigma_{sp}$, $\sigma_{sp}$ and chemical species concentration) at the three different sites (BCN, MSY and MSA).

**Reviewer#4. Specific comment 3. P.8 Line 10: Please, explain more clearly in text the equation term [source] and their units.**

The following sentence has been rephrased as follows aiming to clarify the units of the parameters considered in the MLR equation.

"In this study, we used the $PM_{10}$ source contributions ($\mu g$ $m^{-3}$) as dependent variables in the MLR and the measured $\sigma_{sp}$ and $\sigma_{ap}$ coefficients ($Mm^{-1}$) as independent ones. Thus, the resulting regression coefficients of the model represent the MSE and MAE ($m^2$ $g^{-1}$) of mixed aerosol modes,"

Equations 1 and 2 (as example for MSY site) have been modified in the revised paper for better understanding:

$$\sigma^{\lambda}_{sp,\ PM_{10}} = \left(MSE^{\lambda}_{Secondary\ sulfate} \cdot [Secondary\ sulfate]\right) + \left(MSE^{\lambda}_{Secondary\ nitrate} \cdot [Secondary\ nitrate]\right)$$
$$+ \left(MSE^{\lambda}_{V\text{-}Ni} \cdot [V\text{-}Ni]\right) + \left(MSE^{\lambda}_{Aged\ organics} \cdot [Aged\ organics]\right) + \left(MSE^{\lambda}_{Mineral} \cdot [Mineral]\right)$$
$$+ \left(MSE^{\lambda}_{Aged\ marine} \cdot [Aged\ marine]\right) + \left(MSE^{\lambda}_{Industrial/Traffic} \cdot [Industrial/Traffic]\right) \qquad \text{(Equation 1)}$$

$$\sigma^{\lambda}_{ap,\ PM_{10}} = \left(MAE^{\lambda}_{Secondary\ sulfate} \cdot [Secondary\ sulfate]\right) + \left(MAE^{\lambda}_{Secondary\ nitrate} \cdot [Secondary\ nitrate]\right)$$
$$+ \left(MAE^{\lambda}_{V\text{-}Ni} \cdot [V\text{-}Ni]\right) + \left(MAE^{\lambda}_{Aged\ organics} \cdot [Aged\ organics]\right) + \left(MAE^{\lambda}_{Mineral} \cdot [Mineral]\right)$$
$$+ \left(MAE^{\lambda}_{Aged\ marine} \cdot [Aged\ marine]\right) + \left(MAE^{\lambda}_{Industrial/Traffic} \cdot [Industrial/Traffic]\right) \qquad \text{(Equation 2)}$$

**Reviewer#4. Specific comment 4. P.8 Line 18: the influence of the inlet change (PM2.5 by PM10) may be important, and may affect the measured σsp values and the regression models.**

Authors agree with the referee, the inlet change from PM2.5 to PM10 at MSA may influence the total measured scattering and therefore the MSE obtained in the MLR analysis. Given that a PM2.5 inlet was considered for the major part of the sampling period, the resulting MSE obtained at MSA might be slightly underestimated compared to the values obtained at BCN and MSY, where a PM10 cut-off inlet was installed for the whole period.

As we commented in the paper, an estimation of the influence of the inlet change on the resulting MSE at MSA is difficult to achieve, given the relatively short $\sigma_{sp}$ time series available for the PM10 fraction and the absence of PM2.5 filter sampling, thus preventing performing two different MLR analyses for the two fractions.

The PMF sources that could be more affected by the inlet change are Mineral and Aged marine, given that particles constituting these sources are contained in the coarse fraction. Ripoll et al. (2014) found that mineral matter and sea salt chemical compounds at MSA mostly contribute to the coarse fraction (55% and 3% mass contribution to $PM_{1-10}$, respectively) and poorly contribute to the fine fraction (5% and 1% mass contribution to $PM_1$). This information is supported by the lowest SAE values obtained for Mineral and Aged marine (Table 2 of the paper), evidencing the dominance of coarse particles within these sources. The scattering effect for the rest of the PMF aerosol sources is mainly restricted to the nephelometer wavelengths (from 0.450 to 0.635 µm), given that most of the particles constituting these sources are within the accumulation mode. Therefore, we can consider that the MSE values obtained in $PM_{10}$ for Aged marine and Mineral sources at MSA may be possibly underestimated.

This information has been added to the revised manuscript:

"In this study both $\sigma_{sp}$ and $\sigma_{ap}$ were collected using a $PM_{10}$ cut off inlet, thus guaranteeing uniformity among the performed optical measurements. An exception occurs at MSA, where a $PM_{2.5}$ cut-off inlet was used until March 2014 and then replaced by a $PM_{10}$ inlet. Therefore, a slight overestimation of the

MSE obtained for *Aged marine* and *Mineral* sources at MSA might be expected when aerosol sampling was performed through the PM$_{2.5}$ inlet, given that particles contained in these sources are mainly present in the coarse fraction and significantly contribute to PM$_{1-10}$ mass concentration (Ripoll et al., 2015a). However, an estimation of the influence of the inlet change on the resulting MSE and MAE at MSA is difficult to achieve, given the relatively short $\sigma_{sp}$ and $\sigma_{ap}$ time series available thus preventing performing two different MLR analyses for the two fractions."

**Reviewer#4. Specific comment 5. P.9 and followings. Results section. The authors show average concentrations, and so on, but they do not include standard deviations in values. As example: Line 21. Please, include standard deviations, or an indication of errors, in all values included in Results section and Tables.**

Standard deviations of the average absolute PM10 mass contributions have been added to the revised paper, including Table 1.

**Reviewer#4. Specific comment 6. P.14, Section 3.3.1, Table 2. The scattering Angström exponents (SAE), are computed using the values in Table 2? Or are computed for each 3 λ MSE measurements, and value in Table 2 is the average (include standard deviations)? Also from SSA. What physical interpretation do the SAE negative values have?, and the high value 3.549 at MSA for aged organics? This last value is close to the Rayleigh scattering (=4).**

SAE values are computed from the linear fit of the 3-λ MSE shown in table 2. The following sentence has been modified in the revised paper:
"The source specific scattering Ångström exponents (SAE) were calculated as a linear fit of 3λ MSE in the 450–635 nm range (Table 2). The MSE values used for computing SAE are shown in Table 2."

The SSA values were also obtained by means of the MSE and MAE values shown in table 2. An indication of "Table 2" has been added to the following sentence of the paper:
"The corresponding SSA to each source was computed as the ratio between the source specific MSE and the sum of MSE and MAE (Table 2)."

The negative SAE values obtained for Mineral source are explained by the coarse particles contained within the source, resulting in a higher scattering efficiency at longer wavelengths. Other studies have reported negative SAE values for pure mineral dust particles, for example particles sampled during Saharan dust events (Collaud Cohen et al., 2004; Russell et al., 2010). Negative SAE values were also observed at MSY and MSA sites during Saharan dust events (Ealo et al., 2016).
The high SAE value obtained for the Aged organics source at MSA is explained by the fine particles usually sampled in this site in absence of natural or anthropogenic pollution. It should be noted that

Aged organics is mainly traced by OC (OM), and OM is the major component of PM1 mass concentration (39%) at MSA (Ripoll et al., 2015), showing low contribution in the coarse fraction (PM$_{1-10}$; 14%).

**Reviewer#4. Specific comment 7. Figure 7. Maybe it's better to show the reconstruction only for the independent subset (January-December 2015), and no the period 2004-2014 were the data values are used to the reconstruction model.**

Authors prefer to keep the aforementioned figure in the manuscript given that modeled $\sigma_{sp}$ and $\sigma_{ap}$ coefficients are used for trends study in section 3.6. Moreover, Figure 7 is also relevant for comparison with previous studies (i.e: Ryan et al., 2005; Malm and Halm 2006; Tao et al., 2014; Cheng et al., 2015), where correlations between modelled and measured parameters resulting from applying a MLR analysis are shown and discussed.

[revised manuscript text omitted]

a)

[Figure]

b)

[Figure]

**MSA**
$<PM_{10}> = 9.65\ \mu g/m^3$

**Figure 2**

[Figure]

**Figure 3**

[Figure]

**Figure 4**

[Figure]

**Figure 5**

[Figure]

**Figure 6**

a) BCN

b) MSY

[Figure]

[Figure]

c) MSA

**Figure 7**

[Figure]

**Figure 8**

**Supplementary material of "Impact of aerosol particle sources on optical properties at urban, regional and remote levels in the north-western Mediterranean"**

**Table S1.** Study period considered for the measured parameters ($\sigma_{sp}$, $\sigma_{sp}$ and chemical species concentration) at the three different sites (BCN, MSY and MSA).

| Parameters | BCN | MSY | MSA |
|---|---|---|---|
| $\sigma_{sp}$ | - | 2010-2014 | 2011-2014 |
| $\sigma_{ap}$ | 2009-2014 | 2010-2014 | 2011-2014 |
| Species concentration | 2004-2014 | 2004-2014 | 2010-2014 |

[Figure]

**Figure S1.** Correlation matrix between pairs of aerosol sources obtained by means of the PMF model at a) BCN, b) MSY and c) MSA. The correlation is coded by shape and colour for better visualization, lower correlations are represented by circles and lighter colours whereas higher correlations are represented by ellipses and darker colours.

[Figure]

**Figure S2.** Source chemical profiles obtained by means of the PMF model at MSY and BCN for the period 2004-2014 (Pandolfi et al., 2016).